# The evolutionary modifications of a GoLoco motif in the AGS protein facilitate micromere formation in the sea urchin embryo

**Natsuko Emura†, Florence DM Wavreil†, Annaliese Fries, Mamiko Yajima\***

Department of Molecular Biology, Cellular Biology, Biochemistry, Brown University, Providence, United States

## eLife Assessment

This **important** study presents work on the molecular mechanism driving asymmetric cell division and fate decisions during embryonic development of echinoids. The evidence supporting the claims of the authors is **convincing**. The work will be of interest to developmental biologists and cell biologists working in the field of self-renewal.

**\*For correspondence:**
mamiko_yajima@brown.edu

†These authors contributed equally to this work

**Competing interest:** The authors declare that no competing interests exist.

**Abstract** The evolutionary introduction of asymmetric cell division (ACD) into the developmental program facilitates the formation of a new cell type, contributing to developmental diversity and, eventually, species diversification. The micromere of the sea urchin embryo may serve as one of those examples: an ACD at the 16-cell stage forms micromeres unique to echinoids among echinoderms. We previously reported that a polarity factor, activator of G-protein signaling (AGS), plays a crucial role in micromere formation. However, AGS and its associated ACD factors are present in all echinoderms and across most metazoans. This raises the question of what evolutionary modifications of AGS protein or its surrounding molecular environment contributed to the evolutionary acquisition of micromeres only in echinoids. In this study, we learned that the GoLoco motifs at the AGS C-terminus play critical roles in regulating micromere formation in sea urchin embryos. Further, other echinoderms' AGS or chimeric AGS that contain the C-terminus of AGS orthologs from various organisms showed varied localization and function in micromere formation. In contrast, the sea star or the pencil urchin orthologs of other ACD factors were consistently localized at the vegetal cortex in the sea urchin embryo, suggesting that AGS may be a unique variable factor that facilitates ACD diversity among echinoderms. Consistently, sea urchin AGS appears to facilitate micromere-like cell formation and accelerate the enrichment timing of the germline factor Vasa during early embryogenesis of the pencil urchin, an ancestral type of sea urchin. Based on these observations, we propose that the molecular evolution of a single polarity factor facilitates ACD diversity while preserving the core ACD machinery among echinoderms and beyond during evolution.

## Introduction

Asymmetric cell division (ACD) is a developmental process that facilitates cell fate diversification by distributing fate determinants differently between daughter cells. It is an essential process for multicellular organisms since it creates distinct cell types, leading to different tissues in an organism. For example, in *Drosophila*, embryonic neuroblasts divide asymmetrically to produce apical self-renewing neuroblasts and basal ganglion mother cells (*Bate et al., 1978*; *Doe, 2008*; *Doe et al.,*

*1988*; *Hartenstein and Campos-Ortega, 1984*). In *Caenorhabditis elegans,* the zygote divides asymmetrically to form a large anterior and a small posterior blastomere with distinct cell fates (*Schnabel et al., 1996*; *Sulston et al., 1983*; *Watts et al., 1996*). In mammals, neuroepithelial cells undergo ACD to produce apical self-renewing stem cells as well as basal neural progenitor cells (*Chenn and McConnell, 1995*; *Haydar et al., 2003*; *Konno et al., 2008*; *Noctor et al., 2004*). A set of polarity factors conserved across phyla regulates these highly organized ACD processes. However, the timing and location of such controlled ACD often occur randomly, even within the same phylum, providing uniqueness to the developmental program of each species. Therefore, we hypothesize that drastic changes in the ACD machinery are unnecessary. Instead, a slight modification in the ACD machinery may drive the formation of a new cell type and the change in the developmental program, contributing to species diversification in the process of evolution.

In this study, we use echinoderm embryos as a model system to test this hypothesis. Echinoderms are basal deuterostomes and include sea urchins, sea stars, and sea cucumbers, among others. In the well-studied echinoderm models, the sea urchin and sea star embryos, the first ACD or symmetry break occurs at the eight-cell stage, where a horizontal cell division separates animal and vegetal blastomeres that contribute to ectoderm and endomesoderm lineages, respectively (*Figure 1A*). However, in the next cell cycle at the 16-cell stage, the sea urchin embryo undergoes an apparent unequal cell division, producing four micromeres at the vegetal pole. In contrast, the sea star embryo undergoes a seemingly equal cell division (*Figure 1B*).

Micromere formation in the sea urchin embryo is a highly controlled ACD event since this cell lineage undergoes autonomous cell specification and functions as organizers as soon as it is formed at the 16-cell stage (*Hörstadius, 1928*; *Ransick and Davidson, 1993*). For example, micromeres autonomously divide asymmetrically again to produce large and small micromeres that are committed to two specific lineages, forming skeletogenic cells and the germline, respectively, at the 32-cell stage (*Okazaki, 1975*; *Yajima and Wessel, 2011*). This early segregation of the germline is unique to sea urchins among echinoderms (*Juliano and Wessel, 2009*; *Fresques et al., 2016*). Further, micromeres induce endomesoderm specification (e.g., gastrulation) even when they are placed in the ectopic region of the embryo, such as the animal cap, suggesting they function as a major signaling center in this embryo (*Hörstadius, 1928*; *Ransick and Davidson, 1993*). The removal of sea urchin micromeres results in compromised or delayed endomesoderm development and compensatory upregulation of a germline factor, Vasa, to presumably start over the developmental program (*Ransick and Davidson, 1993*; *Voronina and Wessel, 2006*).

In contrast, other echinoderms undergo minor unequal cell divisions during early embryogenesis, yet they may not be linked to specific cell fate or function. For example, in sea star embryos, the removal of smaller cells does not impact embryonic patterning, and unequal cell divisions do not appear to be linked to specific cell fate regulation or function (*Barone et al., 2022*). Similarly, even in sea urchin embryos, the non-micromere blastomeres formed at the 16-cell stage can change their cell fate in response to external cues, including the signaling from micromeres. Recent studies using single-cell RNA-seq analysis further support these observations by demonstrating the earlier molecular segregation of the micromere lineage, while other cell lineages appear to undergo more regulative development (*Foster et al., 2019*; *Massri et al., 2021*).

Fossil records and phylogenetic tree analysis suggest that sea urchins diverged relatively later from the common ancestor of echinoderms (*Bottjer et al., 2006*; *Wada and Satoh, 1994*). Since micromeres are unique to echinoids (sea urchins, sand dollars, pencil urchins), they are considered to have emerged later during echinoderm diversification, which has dramatically changed the developmental style in the sea urchin embryo (*Emura and Yajima, 2022*). To understand how this unique lineage has emerged during evolution, we previously identified the activator of G-protein signaling (AGS) (Pins in *Drosophila*; LGN in mammals) as a major regulator of micromere formation (*Poon et al., 2019*). AGS is a polarity factor and plays a role in the ACD of many organisms (reviewed by *di Pietro et al., 2016*; *Kotak, 2019*; *Rose and Gönczy, 2014*; *Siller and Doe, 2009*; *Wavreil and Yajima, 2020*; *Yu et al., 2006*). In the sea urchin (*Strongylocentrotus purpuratus;* Sp), SpAGS localizes to the vegetal cortex before and during micromere formation, and its knockdown inhibits micromere formation (*Poon et al., 2019*). On the other hand, in the sea star (*Patiria miniata;* Pm), PmAGS shows no cortical localization nor any significant role in ACD during early embryogenesis. The pencil urchin (*Eucidaris tribuloides;* Et) is an ancestral type of sea urchin that diverged around 252 million years ago, located

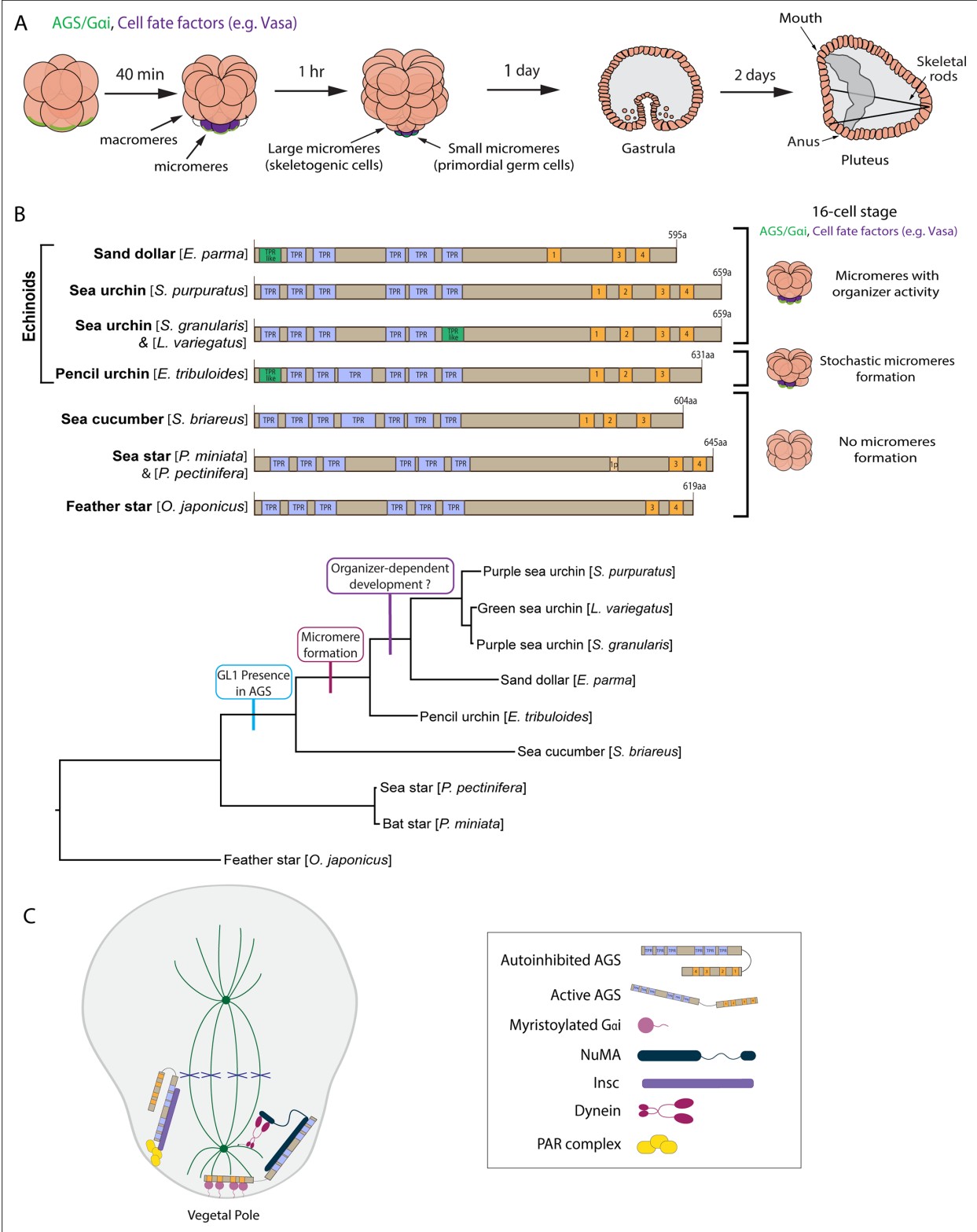

**Figure 1.** The evolutionary modification of the SpAGS protein corresponds to the introduction of micromeres and inductive signaling during echinoid diversification. (**A**) Schema depicting sea urchin embryonic development from eight-cell stage to pluteus. Green represents the colocalization of AGS and Gαi at the vegetal cortex, and purple represents the early segregation of fate factors such as Vasa. (**B**) Comparative diagrams of predicted motifs of each echinoderm AGS protein, based on NCBI blast search results for AGS sequences. Conserved TPR motifs are indicated in blue, and GL motifs in orange. Green shows TPR-like motifs, which contain several amino acid changes. Lighter colors represent partial GL motifs. See **Figure 2—figure**

*Figure 1 continued on next page*

*Figure 1 continued*

**supplement 2** for each echinoderm AGS sequence. The tree depicts SpAGS evolution among echinoderms with the introduction of the GL1 motif and micromeres. (**C**) Working model of AGS mechanism in asymmetric cell division (ACD) based on fly and mammalian models.

between the sea star and the sea urchin on the phylogenetic tree. The pencil urchin embryo exhibits an intermediate developmental program of the sea urchins and sea stars. It stochastically forms 0 to 4 micromere-like cells (*Figure 1B*). In these embryos, EtAGS localizes to the vegetal cortex only when the embryos form micromere-like cells (*Poon et al., 2019*), suggesting a close correlation between cortical AGS localization and micromere-like cell formation.

Furthermore, the introduction of sea urchin AGS into sea star embryos induces random unequal cell divisions by recruiting the spindle poles to the cortex (*Poon et al., 2019*), suggesting that SpAGS facilitates unequal cell divisions even in other echinoderm species. Phylogenetic analysis of AGS orthologs across taxa suggests that AGS orthologs increased the functional motif numbers over the course of evolution, likely allowing additional molecular interactions and mechanisms to modulate ACD in a more nuanced manner in higher-order organisms (*Wavreil and Yajima, 2020*). Supporting this hypothesis, indeed, prior studies suggest that the higher-order mouse AGS ortholog (LGN) can substitute for its fly ortholog (Pins) in *Drosophila* cells, while the basal-order fly Pins cannot substitute its chick ortholog function in chicks, the higher-order organism (*Yu et al., 2003*; *Saadaoui et al., 2017*). These observations led us to hypothesize that the molecular evolution of AGS orthologs drives ACD diversity across taxa, contributing to the developmental diversity within each phylum. In this study, through a series of molecular dissection experiments, we demonstrate that the AGS C-terminus is a variable region and creates its functional diversity in ACD control, facilitating the developmental variations among echinoderms. This study provides insight into how the molecular evolution of a single polarity factor contributes to developmental diversity within each phylum.

## Results

### The N-terminal TPR domain is vital for restricting SpAGS localization and function at the vegetal cortex

AGS consists of two functional domains: the N-terminus contains tetratricopeptide repeats (TPR) and the C-terminus contains G-protein regulatory motifs (GoLoco, GL) (*Bernard et al., 2001*). AGS switches between a closed and open structure based on the intramolecular interaction between the TPR and GL motifs (*Du and Macara, 2004*; *Johnston et al., 2009*; *Nipper et al., 2007*; *Pan et al., 2013*). The TPR motifs regulate protein–protein interaction with various partners such as inscuteable (Insc) for its proper cortical localization or nuclear mitotic apparatus (NuMA) for its microtubule-pulling force generation. In contrast, the GL motifs interact with the heterotrimeric G-protein subunit Gαi for its anchoring to the cortex (*Bowman et al., 2006*; *Culurgioni et al., 2011*; *Culurgioni et al., 2018*; *Du and Macara, 2004*; *Parmentier et al., 2000*; *Wang et al., 2011*; *Yu et al., 2000*). Studies investigating AGS mechanisms in fly and mammals reveal that Pins/LGN (AGS orthologs) generally remain in the autoinhibited form in the cell (*Du and Macara, 2004*; *Johnston et al., 2009*; *Nipper et al., 2007*; *Figure 1C*). At the time of ACD, Insc recruits Pins/LGN to the cortex, which is then established and maintained there through Gαi interaction for the subsequent steps. This Gαi-binding releases Pins/LGN from its autoinhibition and allows it to interact with NuMA, which recruits the motor protein dynein to generate pulling forces on the microtubules and facilitate ACD (*Bowman et al., 2006*; *Culurgioni et al., 2011*; *Izumi et al., 2006*; *Parmentier et al., 2000*; *Schaefer et al., 2001*; *Siller et al., 2006*; *Williams et al., 2014*; *Yu et al., 2000*; *Yuzawa et al., 2011*; *Zhu et al., 2011*).

To test whether sea urchin (*S. purpuratus*; Sp) AGS functions in ACD similarly to its orthologs, we first investigated the role of its N-terminus by constructing a series of GFP-tagged deletion mutants (*Figure 2A*; *Figure 2—figure supplement 1*). AGS-1F is missing the first three TPR motifs, AGS-2F the first four, and AGS-3F the entire TPR domain of SpAGS open-reading frame (ORF). The mRNA for these deletion constructs was co-injected with 2x-mCherry-EMTB, a microtubule marker, to visualize the cell cycle phase, spindle location, and orientation. We counted the number of embryos with vegetal cortical localization and conducted a quantitative analysis by measuring the ratio of vegetal to animal cortical signal intensity at the 16–32-cell stage (*Figure 2B and C*). Embryos injected with full-length SpAGS (Full AGS) or AGS-1F exhibited vegetal cortex-specific localization. In contrast,

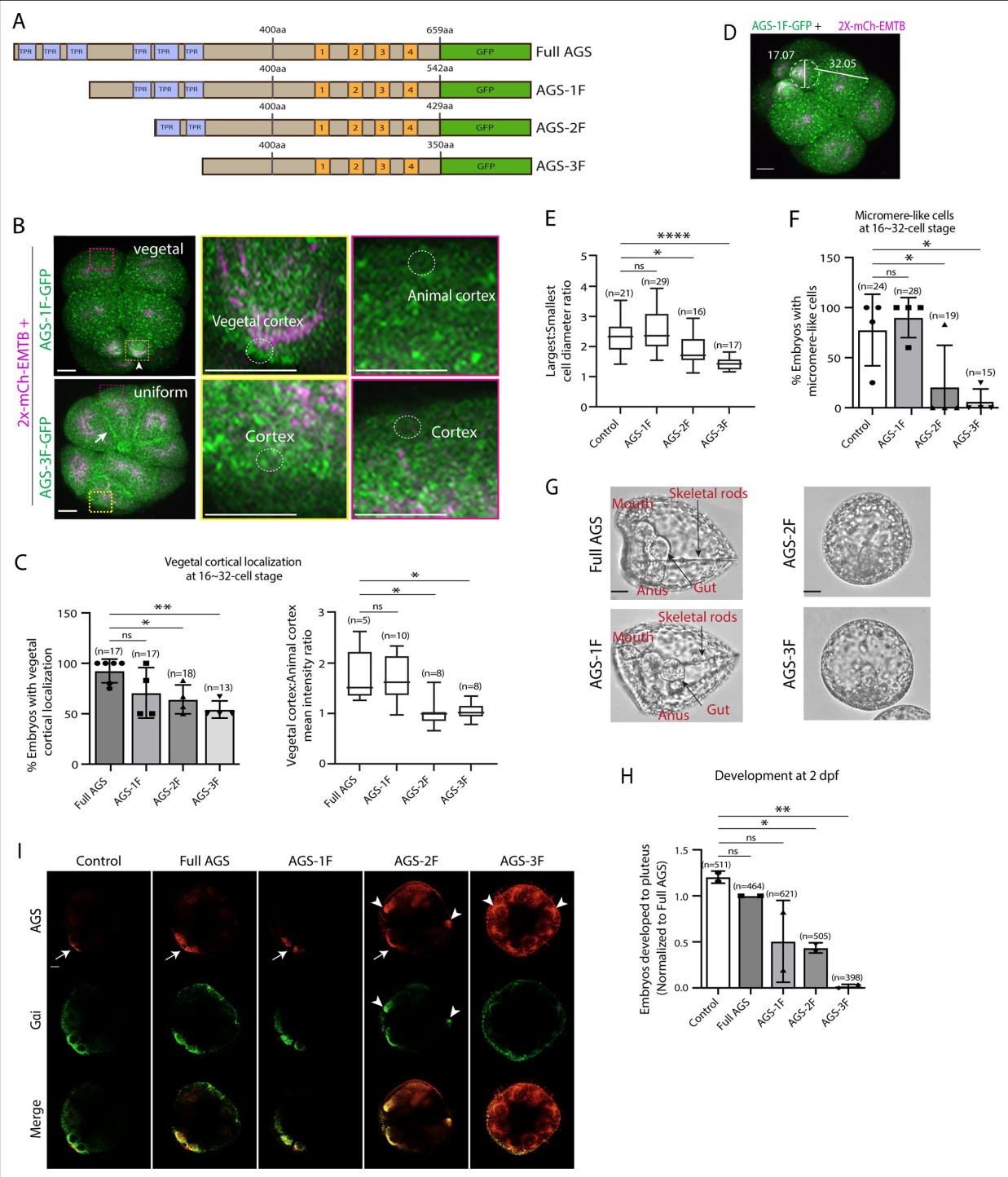

**Figure 2.** The N-terminal TPR domain restricts SpAGS localization and function at the vegetal cortex. (**A**) Design of SpAGS-GFP N-terminal deletion constructs used in this study. TPR motifs are marked in blue, and GL motifs are in orange. (**B**) Representative 2D-projection images of the embryo injected with AGS-1F-GFP or AGS-3F-GFP and 2x-mCherry-EMTB, exhibiting vegetal (upper panel, arrowhead) and uniform (lower panel, arrow) cortical localization. The magnified images next to each panel demonstrate how we measured the mean intensities of the vegetal cortex (yellow squared) and animal cortex (magenta squared) using *ImageJ*. The results of the analysis are summarized in the corresponding graph (**C**). Embryos were injected with 0.15–0.3 µg/µl stock of SpAGS-GFP mRNA and 0.5 µg/µl stock of 2x-mCherry-EMTB mRNA. Z-stack images were taken at 1 µm intervals to cover a layer of the embryo. (**C**) Percentage of the embryos with vegetal cortical localization of SpAGS (left) and the ratio of the vegetal cortex-to-animal cortex mean intensity (right) at 16–32-cell embryos. Statistical analysis was performed against Full AGS by one-way ANOVA. (**D**) Representative 2D-projection confocal image of a 16-cell stage embryo injected with AGS-1F-GFP. The largest cell (macromere) and the smallest cell (micromere) diameters

*Figure 2 continued on next page*

*Figure 2 continued*

were measured using *ImageJ*. Z-stack images were taken at 1 μm intervals to cover a layer of the embryo. (**E**) The diameter ratio of the smallest cell (micromere-like cell) over the largest cell (macromere-like cell) was quantified for the embryos injected with the SpAGS mutants or EMTB-only (control). (**F**) Percentage of the embryos forming micromere-like cells was scored for each SpAGS mutant and EMTB-only (control). 'Micromere formation' is defined as the formation of a group of four cells that are smaller in size and made through a vertical cell division at the vegetal pole at the 16-cell stage. Since none of the AGS-3F-injected embryos formed normal micromeres, 'micromere-like cells' were counted based on their vertical cell division, not relative to their size. Statistical analysis was performed against control by one-way ANOVA. (**G, H**) Brightfield images show the representative phenotypes scored in the corresponding graph (**H**) at 2 dpf. We categorized embryos into three groups, namely, 'full development', with embryos reaching the pluteus stage with complete gut formation and skeleton; 'delayed development', with some gastrulation but no proper skeleton; and 'failed gastrulation'. As many of the abnormal-looking embryos fell into the median of the latter two categories, we scored only the embryos reaching full development in the graph. Control represents embryos injected with a RITC dye only. Statistical analysis was performed against control by one-way ANOVA. (**I**) Single Z-slice confocal imaging was used to focus on the vegetal cortex. Embryos were stained with AGS (orange) and Gαi (green) antibodies. White arrows and arrowheads indicate the signals at the vegetal cortex and ectopic cortical signals, respectively. Images represent over 80% of the embryos observed (n = 30 or larger) per group. n indicates the total number of embryos scored. *p<0.05, **p<0.01, and ****p<0.0001. Each experiment was performed at least three independent times. Error bars represent standard error. Scale bars = 10 μm.

The online version of this article includes the following source data and figure supplement(s) for figure 2:

**Source data 1.** Numerical data for *Figure 2C*.

**Source data 2.** Numerical data for *Figure 2E*.

**Source data 3.** Numerical data for *Figure 2F*.

**Source data 4.** Numerical data for *Figure 2H*.

**Figure supplement 1.** Dissection of SpAGS motifs.

**Figure supplement 2.** Echinoderm AGS sequence alignment.

AGS-2F and AGS-3F showed uniform cortical localization (*Figure 2B and C*). These results suggest that TPR4-6 is necessary for restricting AGS to the vegetal cortex, whereas TPR1-3 appears to play a less critical role in controlling AGS localization.

In the EMTB-only control and the AGS-1F group, micromeres were approximately half the size of the macromeres. In contrast, they were three-quarters the size in the AGS-2F group and almost the same size in the AGS-3F group (*Figure 2D and E*), resulting in failed micromere formation even in the presence of the endogenous SpAGS (*Figure 2F*). In these embryos, we also scored embryonic development at 2 days post fertilization (dpf) when gastrulation occurs. The AGS-1F mutant mostly showed normal development with extended skeletal rods, whereas AGS-2F and AGS-3F dramatically compromised development with incomplete skeleton extension or gut formation (*Figure 2G and H*). Since these N-terminal deletions appear to cause a dominant negative phenotype, we did not knock down endogenous SpAGS in these experiments.

These results suggest that the N-terminal TPR domain is necessary to restrict SpAGS localization at the vegetal cortex. The TPR deletion prevents AGS mutants from maintaining the autoinhibited form. It may thus induce their binding to Gαi at every cortex and compete out the endogenous SpAGS at the vegetal cortex. Notably, Gαi localization was also recruited to the exact ectopic location as AGS-2F and -3F mutants (*Figure 2I*), suggesting that the SpAGS C-terminus is sufficient to control the Gαi localization at the vegetal cortex. Protein sequences of AGS orthologs across echinoderms are almost identical in their N-termini, suggesting that the AGS N-terminus serves as a core functional domain (*Figure 2—figure supplement 2*). In contrast, the AGS C-terminus appears highly variable across echinoderms.

## The C-terminal GL1 motif is essential for SpAGS localization and function in ACD

To test whether a variable AGS C-terminus creates functional diversity in ACD, we made a series of GFP-tagged C-terminus deletion mutants of SpAGS (*Figure 3A*). SpAGS mutants missing GL1 (ΔGL1), GL3 (ΔGL3), or all GL motifs (ΔGL1-4) failed to localize at the vegetal cortex compared to the Full AGS control (*Figure 3B–D*), suggesting that GL1 and GL3 are essential for cortical localization of AGS. Of note, sea urchin embryos randomly show enriched nuclear signal of any fluorescent dye or the GFP signal, likely due to extra space available in the nucleus during early embryogenesis

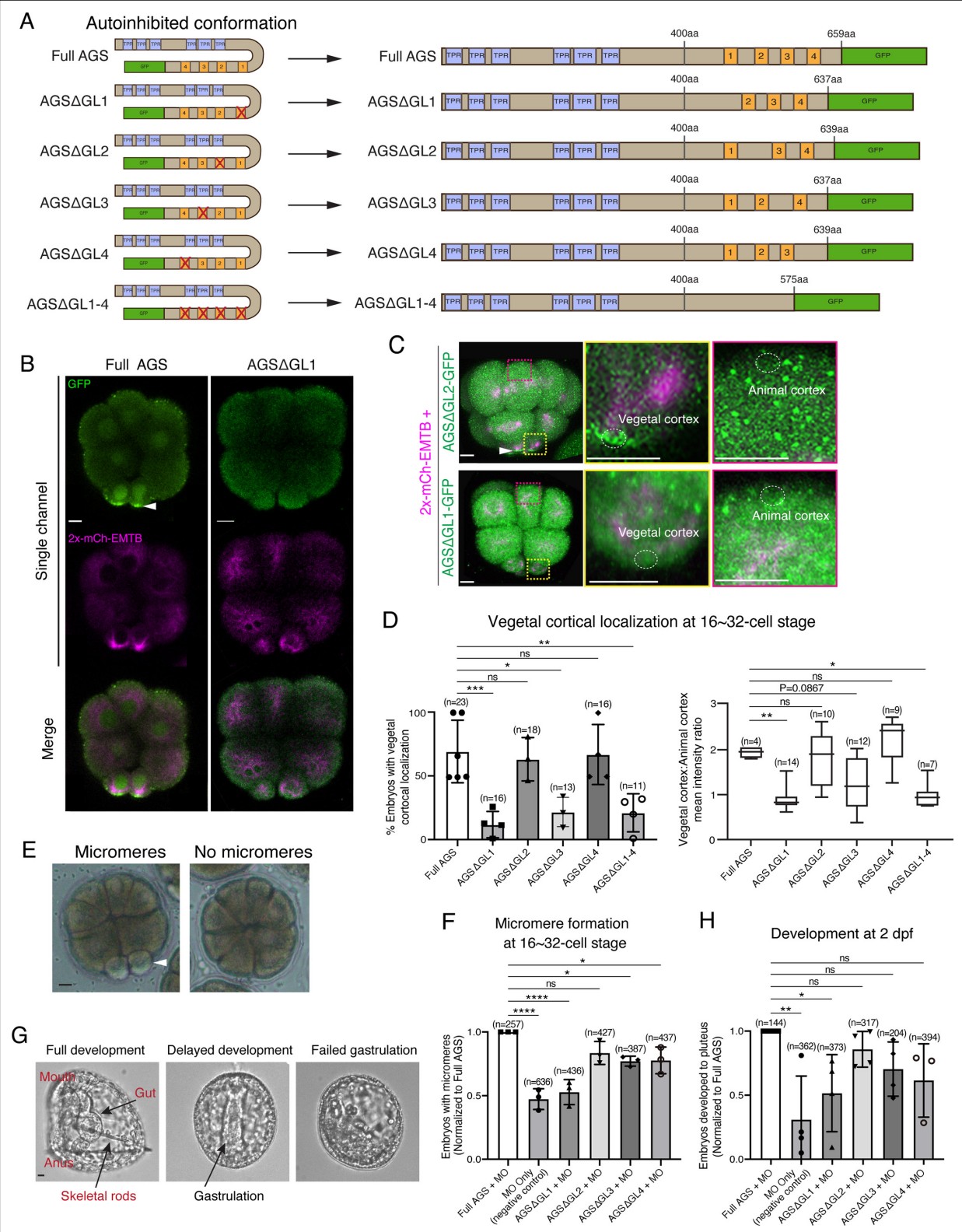

**Figure 3.** GL1 is essential for vegetal cortical recruitment of SpAGS at the 8–16-cell stage of the sea urchin embryo. (**A**) Design of SpAGS-GFP C-terminal deletion mRNAs tested in this study. TPR motifs are marked in blue, and GL motifs are in orange. See *Figure 2—figure supplement 2* for protein sequence. (**B**) Single Z-slice confocal imaging was used to focus on the vegetal cortex. Representative embryos injected with SpAGS-GFP or SpAGSΔGL1-GFP are shown. Embryos were injected with 0.3 μg/μl stock of SpAGS-mutant-GFP mRNA (green) and 0.5 μg/μl stock of 2x-mCherry-

*Figure 3 continued on next page*

*Figure 3 continued*

EMTB mRNA (magenta). The white arrowhead indicates vegetal cortical localization of AGS-GFP. (**C, D**) Representative 2D-projection images of the embryo injected with SpAGS-GFP mRNA and 2x-mCherry-EMTB mRNA, exhibiting vegetal cortical (upper panel, AGSΔGL2, arrowhead) and uniform cytoplasmic (lower panel, AGSΔGL1) localization. The magnified images next to each panel demonstrate how we measured the mean intensities of the vegetal cortex (yellow squared) and animal cortex (magenta squared) using *ImageJ*. The results of the analysis are summarized in the corresponding graph (**D**). Z-stack images were taken at 1 µm intervals to cover a layer of the embryo. Percentage of the embryos that had the GFP signal at the vegetal cortex (left) and the ratio of the vegetal cortex-to-animal cortex mean intensity (right) during the 16–32-cell stage were scored in the graphs. Statistical analysis was performed against Full AGS by one-way ANOVA. (**E, F**) Brightfield images show the representative phenotypes scored in the corresponding graph (**F**) at the 16-cell stage. White arrowhead shows micromeres. Embryos were injected with 0.15 µg/µl stock of SpAGS-GFP mRNAs and 0.75 mM SpAGS MO. The number of embryos forming micromeres was scored and normalized to that of Full AGS in the graph. Statistical analysis was performed by one-way ANOVA. (**G, H**) Brightfield images show the representative phenotypes scored in the corresponding graph (**H**) at 2 dpf. Embryos were injected with 0.15 µg/µl stock of SpAGS-GFP mRNAs and 0.75 mM SpAGS MO. The number of embryos developing to the pluteus stage was scored and normalized to that of Full AGS in the graph. Statistical analysis was performed by one-way ANOVA. n indicates the total number of embryos scored. *p<0.05, **p<0.01, ***p<0.001, and ****p<0.0001. Each experiment was performed at least three independent times. Error bars represent standard error. Scale bars = 10 µm.

The online version of this article includes the following source data for figure 3:

**Source data 1.** Numerical data for *Figure 3D*.

**Source data 2.** Numerical data for *Figure 3F*.

**Source data 3.** Numerical data for *Figure 3H*.

(*Fernandez-Nicolas et al., 2022*). Since the nuclear AGS signal appeared only randomly in some embryos, we did not analyze such signals in this study.

Next, we knocked down endogenous AGS by morpholino antisense oligonucleotides (MO), which was previously validated for the specificity (*Poon et al., 2019*). We tested whether these deletion mutants could rescue micromere formation (*Figure 3E*). The GL1 deletion significantly reduced micromere formation. In contrast, the GL2, GL3, or GL4 deletion showed no or little significant difference in micromere formation compared to the Full AGS control group (*Figure 3F*). Consequently, the GL1 deletion showed significant disruption in embryonic development at 2 dpf, likely due to a lack of micromeres' inductive signaling at the 16-cell stage (*Figure 3G and H*).

These results suggest GL1 is critical for both AGS localization and function at the vegetal cortex for micromere formation. GL3 and GL4 are important for intramolecular binding to the TPR domain in other organisms, which may impact the proper open-close control of AGS protein (*Du and Macara, 2004*; *Johnston et al., 2009*; *Nipper et al., 2007*; *Pan et al., 2013*).

## The position of GL1 is important for SpAGS function in ACD

To determine whether the sequence or positioning of GL1 is essential for the SpAGS function, we next made a series of mutants where the GL motifs were interchanged or replaced (*Figure 4A*). For instance, AGS1111 has all GL motifs replaced with the sequence of GL1, whereas AGS4234 has the sequence of GL1 replaced with that of GL4. Most embryos that formed micromeres displayed vegetal cortical localization for all mutants except for AGS1111 and AGS2222 that severely inhibited micromere formation (*Figure 4B and C*). A small portion (4.14% ± 2.65, n = 170) of AGS2222 embryos formed micromeres. Among the embryos that formed micromeres, AGS2222 always showed vegetal cortical localization, suggesting that AGS localization and micromere formation are closely linked to each other. Additionally, most of the AGS1111 (99.36% ± 0.64, n = 182) and AGS2222 embryos (98.06% ± 1.94, n = 170) displayed ectopic cortical localization around the entire embryo (*Figure 4B and C*). We did not observe this phenotype in the Full AGS control nor in the other two mutants (AGS2134 and AGS4234).

We quantified the function of these AGS mutants in the endogenous AGS-knockdown background. AGS1111 and AGS2222 mutants failed to restore micromere formation at the 16-cell stage, while AGS4234 and AGS2134 mutants rescued micromere formation similarly to Full AGS (*Figure 4D*). Furthermore, Full AGS, AGS2134, and AGS4234 showed comparable development at 2 dpf. In contrast, the AGS1111 and AGS2222 embryos showed disrupted development (*Figure 4E and F*). These results suggest that the GL1 sequence is not essential, but its position is vital. In contrast, the sequence of GL3 or GL4 appears to be critical for restricting AGS localization to the vegetal cortex, perhaps by maintaining the autoinhibited form of AGS through their interaction with the TPR domains.

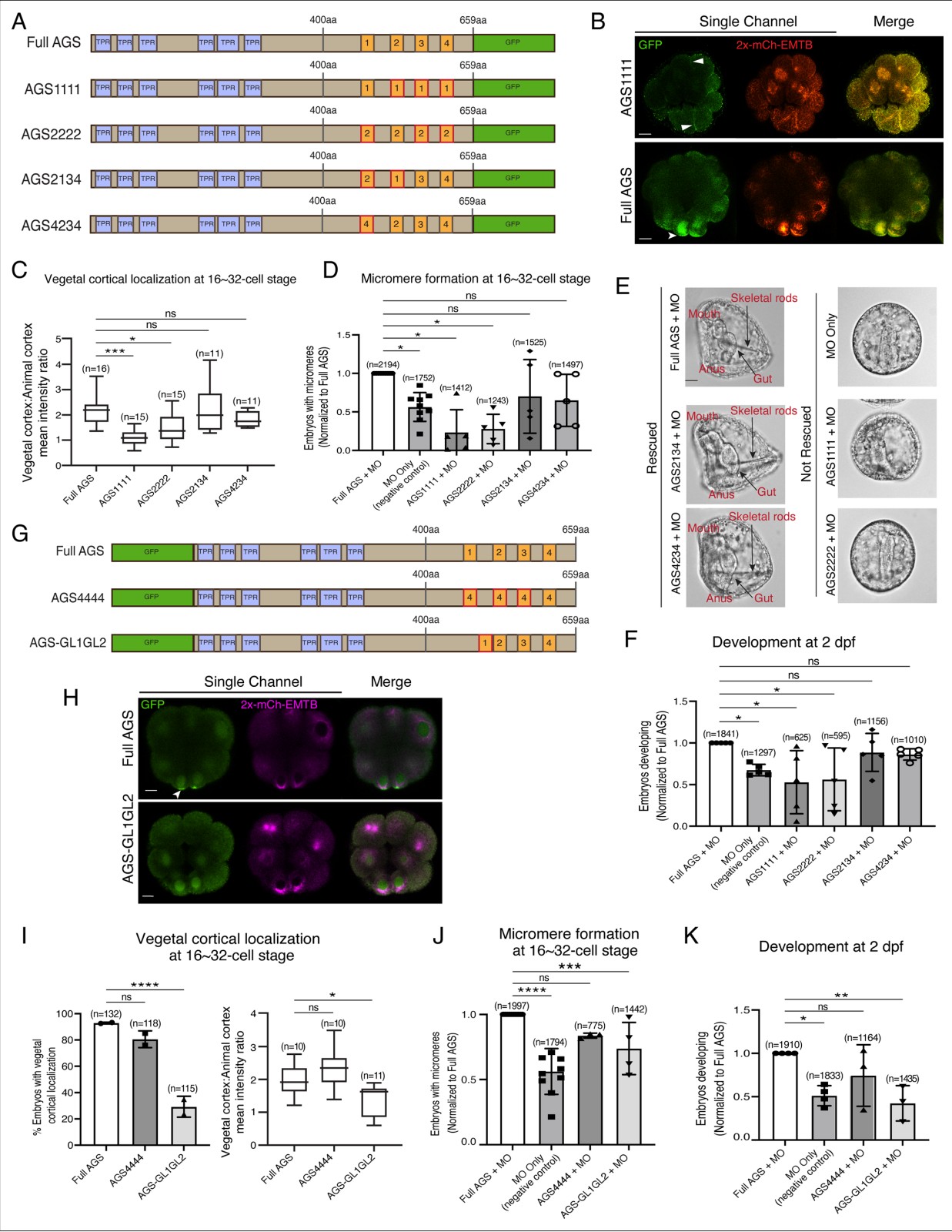

**Figure 4.** The position of GL1 and the sequences of GL3 and GL4 are important for SpAGS localization and function. (**A**) Design of SpAGS-GFP C-terminal mutant constructs tested in this study. TPR motifs are marked in blue, and GL motifs are in orange. Red boxes show interchanged GL motifs. (**B**) Single Z-slice confocal images of sea urchin embryos at the 8–16-cell stage showing localization of SpAGS1111-GFP mutant. Embryos were injected with 0.3 µg/µl stock of SpAGS-mutant GFP mRNAs and 0.25 µg/µl stock of 2x-mCherry-EMTB mRNA. White arrowheads indicate vegetal cortical localization

*Figure 4 continued on next page*

*Figure 4 continued*

of AGS-GFP proteins and ectopic localization of AGS1111 mutant. (**C**) The ratio of the vegetal cortex-to-animal cortex mean intensity in 16–32-cell embryos. Statistical analysis was performed against Full AGS by one-way ANOVA. (**D–F**) Embryos were injected with 0.15 µg/µl stock of SpAGS-GFP mRNAs and 0.75 mM SpAGS MO. The number of embryos making micromeres (**D**) and developing to gastrula or pluteus stage (**F**) was scored and normalized to that of the Full AGS control group. Brightfield images (**E**) show the representative phenotypes scored in the corresponding graph (**F**) at 2 dpf. Of note, AGS1111 and AGS2222 mutants caused substantial toxicity, degrading many embryos by 2 dpf and resulting in inconsistent scoring. Thus, we scored embryos reaching the pluteus stage, which revealed delayed development in this analysis. Statistical analysis was performed by one-way ANOVA. (**G**) Design of GFP-SpAGS C-terminal mutant constructs tested in this study. In AGS4444, we replaced all GL motifs with GL4. In AGS-GL1GL2, GL1 is shifted adjacent to GL2. TPR motifs are marked in blue, and GL motifs are in orange. Red boxes show modified GL motifs. (**H**) Single Z-slice confocal images of sea urchin embryos at the 8–16-cell stage showing localization of GFP-SpAGS-GL1GL2 mutant. Embryos were injected with 0.3 µg/µl stock of GFP-SpAGS mRNA and 0.25 µg/µl stock of 2x-mCherry-EMTB mRNA. The white arrowhead indicates the vegetal cortical localization of GFP-AGS. (**I**) Percentage of the embryos with vegetal cortical localization of SpAGS mutants (left) and the ratio of the vegetal cortex-to-animal cortex mean intensity (right) in 16–32-cell embryos. Statistical analysis was performed against Full AGS by one-way ANOVA. (**J, K**) Embryos were injected with 0.15 µg/µl stock of GFP-SpAGS mRNAs and 0.75 mM SpAGS MO. The number of embryos making micromeres (**J**) and developing to gastrula or pluteus stage (**K**) was scored and normalized to that of the Full AGS. Statistical analysis was performed by one-way ANOVA. n indicates the total number of embryos scored. *$p<0.05$, **$p<0.01$, ***$p<0.001$, and ****$p<0.0001$. Each experiment was performed at least two independent times. Error bars represent standard error. Scale bars = 10 µm.

The online version of this article includes the following source data for figure 4:

**Source data 1.** Numerical data for *Figure 4C*.

**Source data 2.** Numerical data for *Figure 4D*.

**Source data 3.** Numerical data for *Figure 4F*.

**Source data 4.** Numerical data for *Figure 4I*.

**Source data 5.** Numerical data for *Figure 4J*.

**Source data 6.** Numerical data for *Figure 4K*.

AGS1111 and AGS2222 mutants were thus unable to sustain a closed/inactive state, resulting in a constitutively active form all around the cortex. These constitutive active AGS mutants likely further randomized the embryonic polarity in the absence of endogenous AGS, resulting in even worse developmental outcomes than the negative control (*Figure 4F*).

To test this model further, we made two additional SpAGS mutants, AGS4444 and AGS-GL1GL2 (*Figure 4G*). AGS4444 localized properly at the vegetal cortex, whereas AGS-GL1GL2 showed significantly fewer embryos with vegetal cortical localization (*Figure 4H and I*). Furthermore, AGS-GL1GL2 showed impaired function in micromere formation and development compared to Full AGS control (*Figure 4J and K*). On the other hand, AGS4444 showed no significant difference in the proportion of embryos with micromeres at the 16-cell stage and normal development at 2 dpf compared to the Full AGS control. These results further support the hypothesis that GL3 and GL4 are essential for maintaining SpAGS in a closed conformation. Additionally, the position of GL1 is critical for SpAGS localization and function.

## The molecular evolution of the AGS C-terminus facilitates the ACD diversity among AGS orthologs

To understand if/how SpAGS functions uniquely compared to other echinoderm AGS orthologs, we cloned sea star (*P. miniata*; Pm) and pencil urchin (*E. tribuloides*; Et) AGS into the GFP-reporter construct (*Figure 5A*) and introduced them into the sea urchin zygotes. EtAGS showed no significant difference in localization and function compared to the SpAGS control, whereas PmAGS failed in vegetal cortical localization and micromere formation and function (*Figure 5B–E*). Hence, PmAGS is incapable of inducing micromere formation.

Since the N-terminal sequences of SpAGS and PmAGS are almost identical (*Figure 2—figure supplement 2*), we hypothesize that the variable C-terminus made a difference in AGS localization and function at the vegetal cortex. To test this hypothesis, we constructed a series of chimeric SpAGS mutants that replaced its C-terminus with that of other AGS orthologs (*Figure 5F*). These AGS orthologs include human LGN, *Drosophila* (Dm) Pins, and EtAGS, which are all involved in ACD (*Gönczy, 2008*; *Schaefer et al., 2000*; *Wavreil and Yajima, 2020*; *Zhu et al., 2011*, 2011b), as well as human AGS3 and PmAGS, neither of which is involved in ACD (*Saadaoui et al., 2017*).

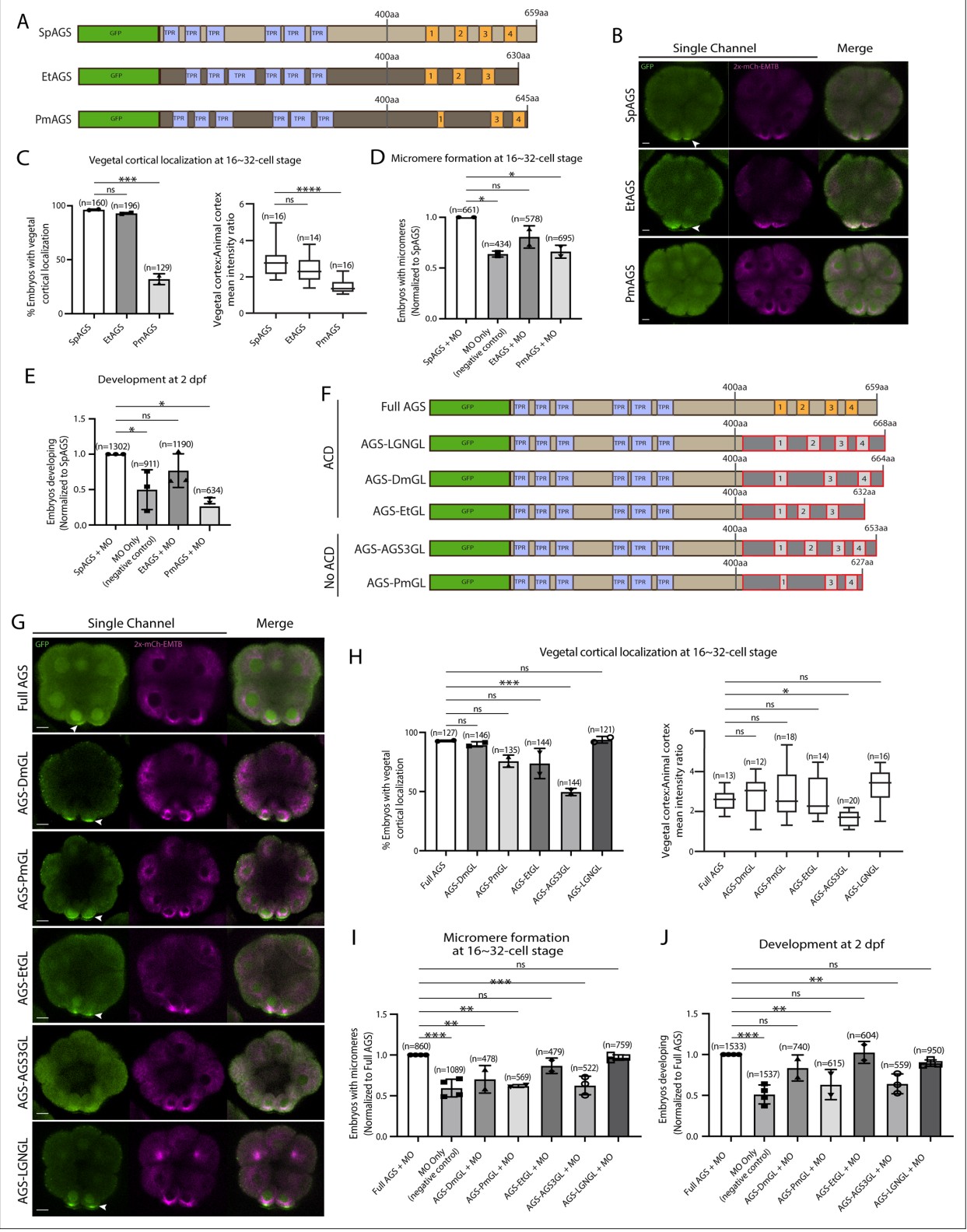

**Figure 5.** Molecular evolution of AGS C-terminus facilitates micromere formation. (**A**) Design of GFP-AGS constructs from three different species tested in this study, namely, *S. purpuratus* (Sp), *E. tribuloides* (Et), and *P. miniata* (Pm). TPR motifs are marked in blue, and GL motifs are in orange. (**B**) Single Z-slice confocal images of sea urchin embryos at 16-cell stage showing localization of each GFP-AGS. Embryos were injected with 0.3 μg/μl stock of GFP-AGS mRNAs and 0.25 μg/μl stock of 2x-mCherry-EMTB mRNA. The white arrowhead indicates vegetal cortical localization of GFP-AGS. (**C**) Percentage of embryos with vegetal cortical localization of GFP-AGS (left) and the ratio of the vegetal cortex-to-animal cortex mean intensity (right) in

*Figure 5 continued on next page*

*Figure 5 continued*

16–32-cell embryos. Statistical analysis was performed against SpAGS by one-way ANOVA. (**D, E**) Embryos were injected with 0.15 µg/µl stock of GFP-AGS mRNAs and 0.75 mM SpAGS MO. The number of embryos making micromeres (**D**) and developing to pluteus stage (**E**) was scored and normalized to that of the Full AGS. Statistical analysis was performed by one-way ANOVA. (**F**) Design of GFP-AGS C-terminal chimeric mutant constructs tested in this study. TPR motifs are marked in blue, and GL motifs are in orange. The brown section shows the SpAGS portion, and the red and dark gray boxes show the non-sea urchin (non-SpAGS) C-terminal sequence introduced. Protein sequences used include *Drosophila* Pins (Dm), *P. miniata* AGS (Pm), *E. tribuloides* AGS (Et), *H. sapiens* AGS3 (AGS3), and *H. sapiens* LGN (LGN). (**G**) Single Z-slice confocal images of sea urchin embryos at the 8–16-cell stage showing localization of each GFP-AGS. Embryos were injected with 0.3 µg/µl stock of GFP-AGS mRNA and 0.25 µg/µl stock of 2x-mCherry-EMTB mRNA. The white arrowheads indicate vegetal cortical localization of GFP-AGS. (**H**) Percentage of the embryos with vegetal cortical localization of GFP-AGS chimeric mutants (left) and the ratio of the vegetal cortex-to-animal cortex mean intensity (right) in 16–32-cell embryos. Statistical analysis was performed against Full AGS by one-way ANOVA. (**I, J**) Embryos were injected with 0.15 µg/µl stock of GFP-AGS mRNAs and 0.75 mM SpAGS MO. The number of embryos making micromeres (**I**) and developing to gastrula or pluteus stage (**J**) was scored and normalized to that of the Full AGS. Statistical analysis was performed by one-way ANOVA. n indicates the total number of embryos scored. *p<0.05, **p<0.01, ***p<0.001, and ****p<0.0001. Each experiment was performed at least two independent times. Error bars represent standard error. Scale bars = 10 µm.

The online version of this article includes the following source data and figure supplement(s) for figure 5:

**Source data 1.** Numerical data for *Figure 5C*.

**Source data 2.** Numerical data for *Figure 5D*.

**Source data 3.** Numerical data for *Figure 5E*.

**Source data 4.** Numerical data for *Figure 5H*.

**Source data 5.** Numerical data for *Figure 5I*.

**Source data 6.** Numerical data for *Figure 4J*.

**Figure supplement 1.** SbAGS does not fully localize at the vegetal cortex.

**Figure supplement 1—source data 1.** Numerical data for *Figure 5—figure supplement 1D*.

**Figure supplement 2.** The linker domain and GL2-GL3 regions are important for AGS localization and function.

**Figure supplement 2—source data 1.** Numerical data for *Figure 5—figure supplement 2D*.

**Figure supplement 2—source data 2.** Numerical data for *Figure 5—figure supplement 2E*.

**Figure supplement 2—source data 3.** Numerical data for *Figure 5—figure supplement 2F*.

**Figure supplement 2—source data 4.** Numerical data for *Figure 5—figure supplement 2I*.

**Figure supplement 2—source data 5.** Numerical data for *Figure 5—figure supplement 2J*.

**Figure supplement 2—source data 6.** Numerical data for *Figure 5—figure supplement 2K*.

**Figure supplement 3.** Alignment of C-terminus GoLoco domain sequences used for chimeric mutants.

The chimeras of ACD-facilitating orthologs (EtGL, LGNGL, DmGL) showed no significant difference in the vegetal cortical localization and micromere function compared to the SpAGS control (*Figure 5G–J*). In contrast, chimeras of non-ACD-facilitators (AGS3GL and PmGL) failed in micromere formation and function. These results suggest that the AGS C-terminus creates ACD diversity by primarily reflecting the original function of each ortholog in the host species. Of note, *Drosophila* Pins chimera (DmGL) showed reduced micromere formation (*Figure 5I*), which may be due to fewer functional domains with decreased efficacy in the higher-order organism (*Wavreil and Yajima, 2020*).

To test this point further, we extended our investigation to the sea cucumber AGS (SbAGS). Sea cucumber embryos do not form micromeres, yet SbAGS has a predicted GL1 motif. However, we noticed that when comparing to the echinoids' AGS orthologs, the GL1 motif of SbAGS is quite different in sequence, and its location is closer to the GL2 motif due to an 8 amino acid deletion (*Figure 5—figure supplement 1A and B*). We hypothesized that SbAGS might not have a full recruitment activity due to these alterations in its GL1 motif. To test this hypothesis, we synthesized the SbAGS sequence and fused it with the 2x-GFP reporter, which provided the same yet brighter signal than 1xGFP for SpAGS (*Figure 5—figure supplement 1C*). We then introduced the SbAGS mRNA into sea urchin (Sp) embryos. We took this synthetic approach since we were unable to obtain sea cucumber embryos for this study. Sea cucumbers are an emerging yet still less established model for experimental biology (*Perillo et al., 2021*). In the resultant sea urchin embryos at the 16-cell stage, the SbAGS signal appeared slightly enriched on the asters and in the cytoplasm of micromeres. However, its signal at the vegetal cortex was significantly reduced compared to the 2x-GFP-SpAGS control (*Figure 5—figure supplement 1C and D*). These results suggest that SbAGS is less recruited

to the vegetal cortex. SbAGS also lacks the GL4 motif, which likely further lessens its function in ACD in sea cucumber embryos. However, further validations using real sea cucumber samples will be essential in the future.

Another unexpected result obtained in this study is that AGS-PmGL showed cortical localization yet still failed to facilitate ACD (*Figure 5G–I*). This result suggests that vegetal cortical localization of AGS does not automatically grant its function in ACD. Perhaps other elements of SpAGS outside of its C-terminus can drive its vegetal cortical localization. One possibility involves the linker region. Indeed, it has been reported that Aurora A phosphorylates the serine in the linker region of Pins, which recruits Pins to the cortex and partially controls the spindle orientation in *Drosophila* (*Johnston et al., 2009*). The phosphorylation site prediction algorithm GPS 6.0 (*Chen et al., 2023*) reveals that SpAGS and EtAGS have the predicted Aurora A phosphorylation site within the linker region, while PmAGS does not (*Figure 5—figure supplement 2A*). To test if this serine is essential for SpAGS localization, we mutated it to alanine (AGS-S389A in *Figure 5—figure supplement 2*). Compared to the Full AGS control, the mutant AGS-S389A showed reduced vegetal cortical localization (*Figure 5—figure supplement 2C and D*) and ACD function (*Figure 5—figure supplement 2E and F*).

The GPS 6.0 predicts that replacing the four amino acids of PmAGS with that of SpAGS could introduce the Aurora A phosphorylation site in the linker region (*Figure 5—figure supplement 2A*, red amino acids). Therefore, we mutated these amino acids to make the PmAGS-SpLinker mutant (*Figure 5—figure supplement 2G*). However, this mutant did not restore any cortical localization nor proper function in ACD (*Figure 5—figure supplement 2H–K*). This result suggests that restoring the predicted Aurora A phosphorylation site is insufficient to induce the cortical localization of AGS. The SpAGS linker region contains multiple predicted phosphorylation sites other than the Aurora A site (*Figure 5—figure supplement 2A*). Therefore, other sites in the linker domain might contribute to AGS recruitment to the vegetal cortex in sea urchin embryos, which needs to be further investigated in the future.

Lastly, in humans, it is proposed that the interdomain sequence between GL2 and GL3 is important for intramolecular interaction with TPR through phosphorylation, mediating the autoinhibitory state of LGN differently from that of AGS3 (*Takayanagi et al., 2019*). To test the importance of the interdomain sequence and its possible phosphorylation, we made mutants targeting the residues unique to the AGS3 GL2-GL3 interdomain region (green and red residues in *Figure 5—figure supplement 3*). One mutant replaces the three serine residues to alanine (AGS3GL-3S/A). Another mutant (AGS3GL-GL2GL3) has five mutations to replace the AGS3 amino acids with that of LGN (S549N, G573D, N578D, Y583C, S585G). Consistent with our hypothesis, the chimera replaced with the LGN residues (AGS3GL-GL2GL3) gained the proper localization and function, while the chimera with serine alterations (AGS3GL-3S/A) failed to function in ACD (*Figure 5—figure supplement 2C–F*). These results suggest that specific amino acid residues within the GL3 motif and between GL2-GL3 are critical, likely mediating interaction with TPR domains and the autoinhibited state of AGS. This result aligns with the earlier results of AGS1111 and AGS2222, which failed in ACD. On the other hand, potential serine phosphorylation between GL2-GL3 motifs appears to be irrelevant to the AGS function.

Overall, we conclude that the variable C-terminus of AGS orthologs primarily facilitates ACD diversity. At the same time, the N-terminus and the linker region of AGS appear to help mediate its autoinhibited state or recruitment, which regulates its cortical localization (summary diagrams in *Figure 6*).

## SpAGS is a dominant factor for micromere formation

Since AGS is a part of the conserved ACD machinery, we next sought to understand how dominant SpAGS is for micromere formation. The other conserved ACD factors include Insc, Discs large (Dlg), NuMA, and Par3 (*Figure 1C*). Insc controls cortical localization of Pins and LGN in flies and humans, respectively (*Schaefer et al., 2000*; *Williams et al., 2014*; *Yu et al., 2000*; *Culurgioni et al., 2011*; *Culurgioni et al., 2018*). Dlg appears to bind to the phosphorylated linker domain of Pins, which recruits microtubules to the cortex in flies (*Johnston et al., 2009*; *Siegrist and Doe, 2005*). NuMA (Mud in *Drosophila*) interacts with LGN/Pins to generate pulling forces on the microtubules in humans and flies. Par3 (Baz in *Drosophila*) is part of the PAR complex with Par6 and aPKC and binds to Insc to help localize LGN/Pins at the cortex (*Culurgioni et al., 2011*; *Parmentier et al., 2000*; *Schaefer et al., 2000*; *Schober et al., 1999*; *Wodarz et al., 2000*; *Yu et al., 2000*).

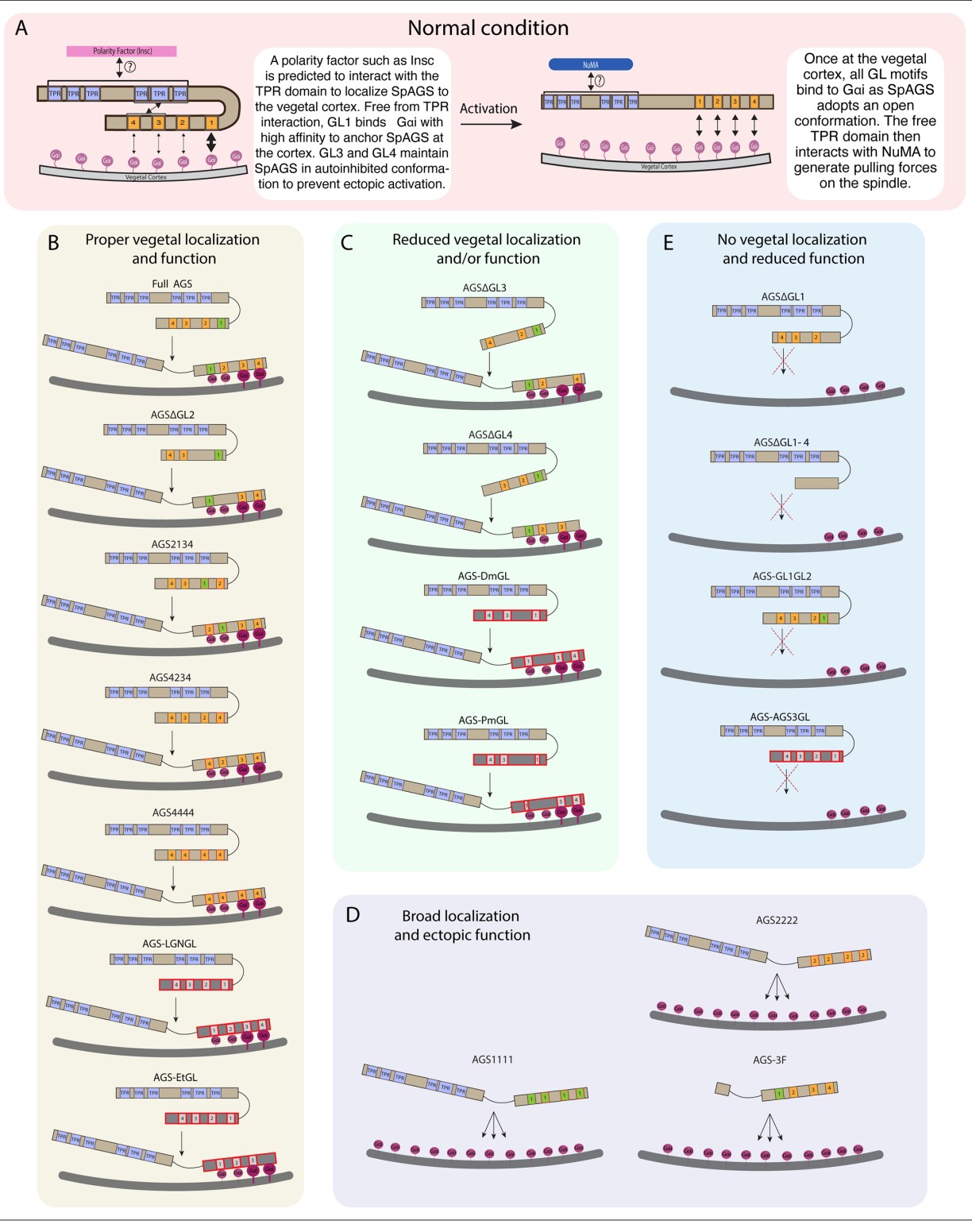

**Figure 6.** Summary diagrams of SpAGS dissection experiments. (**A**) A model for the mechanism of SpAGS localization and function at the vegetal cortex. In a closed conformation, GL1 is critical for SpAGS recruitment and anchoring at the cortex through Gαi binding, while GL3 and GL4 maintain the autoinhibition. The TPR domain is hypothesized to interact with a polarity factor such as Insc to restrict SpAGS localization to the vegetal cortex. Upon Gαi binding, SpAGS adopts an open conformation, allowing all four GLs to bind to Gαi and the TPR domain to interact with NuMA for force generation on the astral microtubules. (**B**) A series of mutants that showed normal vegetal localization and functions. The position of GL1 is a more determining factor since mutants with GL1 replaced with other GL sequences localized and functioned properly. (**C**) A series of mutants that showed

*Figure 6 continued on next page*

*Figure 6 continued*

a reduced vegetal localization and/or function. The GL3 and GL4 are necessary to regulate AGS localization and function, likely by mediating its autoinhibitory mechanism through their binding to TPRs. Furthermore, AGS-DmGL and -PmGL were categorized in this group due to the reduced number of GL motifs. (**D**) A series of mutants that showed broad AGS localization and ectopic function. The TPR domain is critical for restricting AGS localization at the vegetal cortex since its removal spreads the AGS signal around all cortices. The sequences of GL3 and GL4 are also crucial for the SpAGS function. (**E**) A series of mutants that showed neither vegetal localization nor function. Removing or displacing GL1 led to significant disturbances in AGS localization and function, suggesting that having a GL motif at this specific position is critical for AGS interaction with Gαi and its anchoring to the cortex.

We cloned the sea urchin orthologs of these ACD factors and tagged each ORF with a GFP reporter. GFP live imaging or immunofluorescence of these ACD factors showed the highest signal enrichment at the vegetal cortex during or upon micromere formation, as well as on the spindle of all blastomeres to some extent (*Figure 7A*, *Figure 7—figure supplement 1*; *Poon et al., 2019*). Furthermore, we tested for physical interaction by performing a proximity ligation assay (PLA) for AGS and ACD factors (Insc, NuMA, Dlg). Many of these ACD factors are primarily enriched at the vegetal cortex and secondarily localized on the spindle area of all cells (*Figure 7A*; *Poon et al., 2019*). In this study, however, the PLA signal was restricted to the vegetal cortex with the current resolution of the system. The results suggest these ACD factors physically interact with AGS primarily at the vegetal cortex (*Figure 7B*). Hence, the core ACD machinery is present at the vegetal cortex and interacts with AGS. We also observed AGS interacting with a fate determinant, Vasa, that is known to be enriched in micromeres at the vegetal cortex (*Figure 7B*; *Voronina and Wessel, 2006*). These results indicate that AGS may recruit both ACD factors and fate determinants to the vegetal cortex, directly facilitating rapid lineage segregation of micromeres.

Consistent with this observation, SpAGS knockdown reduced the signal enrichment of ACD factors and another fate determinant of micromeres, β-catenin (*Logan et al., 1999*; *Figure 7C–H*). In our previous study (*Poon et al., 2019*), we also identified that SpAGS recruits the spindle poles to every cortex when overexpressed (*Figure 7—figure supplement 2A*, arrows). Similarly, SpAGS at least partially recruits its partner proteins to the ectopic cortical region, which we never observed in the control group (*Figure 7—figure supplement 2B and C*, arrows). These results support the idea that SpAGS directly recruits the molecules essential for micromere lineage segregation. Indeed, in situ hybridization (ISH) analysis suggests that the downstream genes regulated by micromere signaling, such as endomesoderm marker genes (*wnt8*, *foxa*, *blimp1b,* and *endo16*), decreased their expression territories in the AGS-knockdown embryos (*Figure 8*). In contrast, ectoderm (*foxq2*) and skeletogenic mesoderm (*ets1*, *alx1*, *tbr1*, and *sm50*) marker genes showed no significant change in their expressions by AGS knockdown. Overall, these results suggest that SpAGS directly recruits multiple ACD factors and fate determinants necessary for micromere formation and functions as an organizer, facilitating the downstream gene expressions necessary for endomesoderm specification.

## AGS serves as a variable factor in the conserved ACD machinery

AGS shows a variable C-terminal domain and appears to be a primary factor facilitating ACD diversity. However, is AGS the only variable factor among the ACD machinery? To test this question, we cloned and injected orthologs of other ACD factors, such as Insc and Dlg, from pencil urchins (Et) or sea stars (Pm) into sea urchins. Both Insc and Dlg possess relatively conserved functional domains among the three echinoderms with an extra PDZ domain present in PmDlg (*Figure 9A and B*; *Figure 8—figure supplements 1 and 2*). Importantly, these Pm and Et ACD factors showed cortical localization at the vegetal cortex in the sea urchin embryo (*Figure 9C–F*). These results are in stark contrast to the earlier results of Pm/Et AGS, which showed varied localization and function in ACD. Therefore, Insc and Dlg might not be the significant variable factors controlling ACD.

To determine how dominantly SpAGS facilitates ACD diversity, we introduced SpAGS, EtAGS, or PmAGS into the pencil urchin, an ancestral type of sea urchin, and compared their function. We co-introduced Vasa-mCherry to identify the development of the germline, which is one of the micromere descendants. Pencil urchin embryos typically form 0–4 micromere-like cells randomly (*Figure 10A*). Notably, only SpAGS injection increased the formation of micromere-like cells in the resultant pencil urchin embryos. In contrast, EtAGS and PmAGS showed no significant difference from the negative

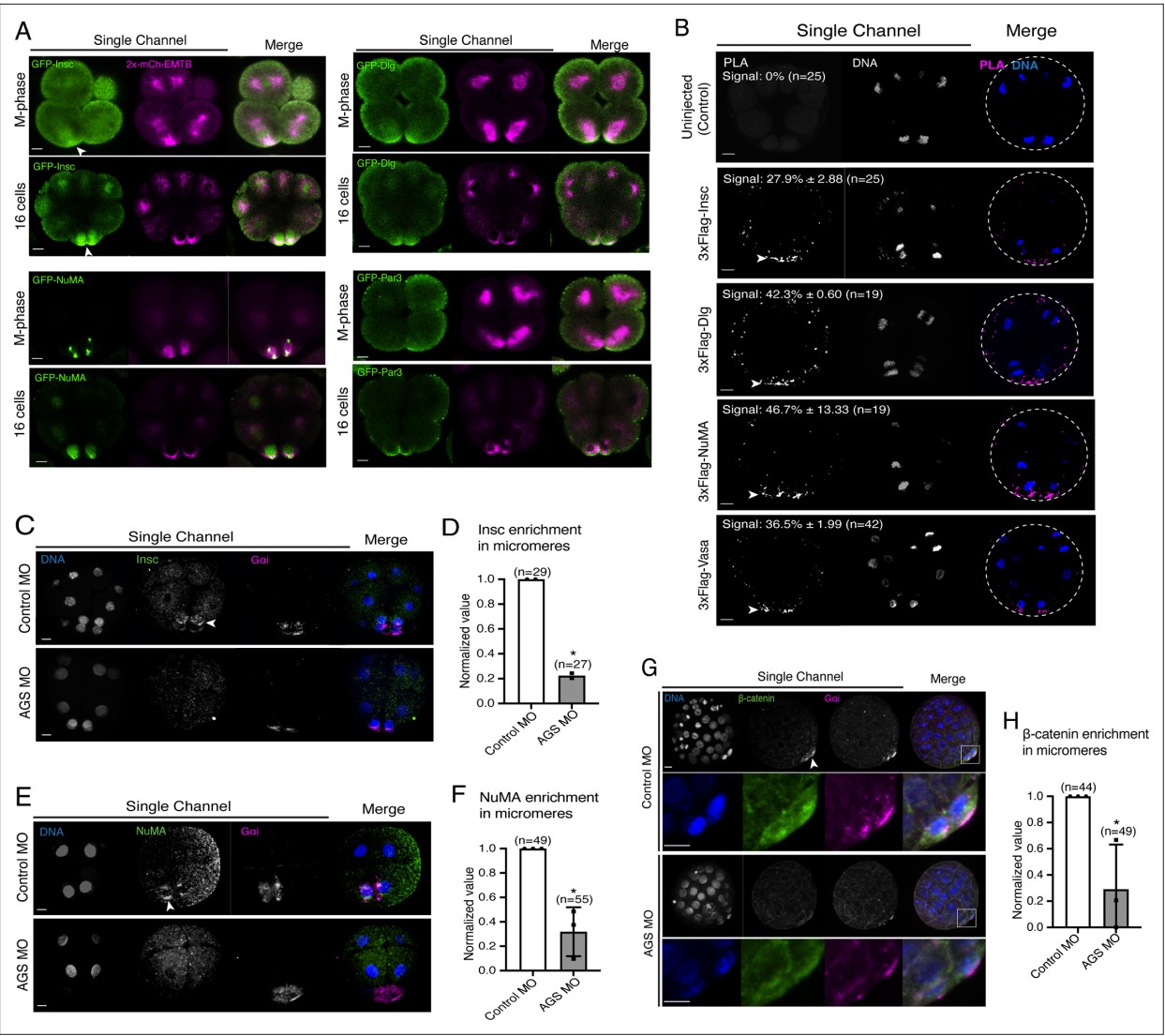

**Figure 7.** SpAGS is critical for the proper localization of asymmetric cell division (ACD) factors and fate determinants. (**A**) Single Z-slice confocal imaging was used to focus on the vegetal cortex. Representative images of embryos during the metaphase and at the 16-cell stage show localization of each GFP-ACD factor, SpInsc, SpDlg, SpNuMA, and SpPar3. Embryos were injected with 0.5 µg/µl stock of GFP-ACD factor mRNAs and 0.25 µg/µl stock of 2x-mCherry-EMTB mRNA. White arrowheads indicate vegetal cortical localization of GFP constructs. Images represent over 80% of the embryos observed (n = 30 or larger) per group across at least two independent cycles of experiments. (**B**) Single Z-slice confocal images of sea urchin embryos at the 8–16-cell stage showing the signals at the vegetal cortex by proximity ligation assay (PLA) assay with Flag and AGS antibodies. Embryos were injected with 0.3–1 µg/µl stock of 3xFlag-ACD factor mRNA. White arrowheads indicate the colocalization of AGS and another ACD factor at the vegetal cortex. The average % of the 8-cell and 8–16-cell embryos with the PLA signal across two independent cycles of experiments is indicated in each image. All embryos were scored independently of the angle since it was hard to identify the angle at the 8-cell stage. (**C–F**) Representative 2D-projection images of the embryo stained with Insc (**C**), NuMA (**E**), and β-catenin (**G**) antibodies (green) by immunofluorescence. Embryos were stained with Gαi antibody (magenta) and Hoechst dye (blue) as well. Z-stack images were taken at 1 µm intervals to cover a layer of the embryo. White arrowheads indicate the signal in micromeres. Embryos were injected with 0.75 mM control MO or 0.75 mM SpAGS MO. The number of embryos showing the localization of Insc (**D**), NuMA (**F**), and β-catenin (**H**) in micromeres was scored and normalized to that of the control MO. Statistical analysis was performed by *t*-test. n indicates the total number of embryos scored. *p<0.05. Each experiment was performed at least two independent times. Error bars represent standard error. Scale bars = 10 µm.

The online version of this article includes the following source data and figure supplement(s) for figure 7:

**Source data 1.** Numerical data for *Figure 7D*.

**Source data 2.** Numerical data for *Figure 7F*.

**Source data 3.** Numerical data for *Figure 7H*.

**Figure supplement 1.** Insc protein expression during embryonic development.

*Figure 7 continued on next page*

*Figure 7 continued*

**Figure supplement 1—source data 1.** Original blots with labels for *Figure 7—figure supplement 1B*.

**Figure supplement 1—source data 2.** Original blot images for *Figure 7–Supplement 1C*.

**Figure supplement 1—source data 3.** The PDF file of original blots with labels for *Figure 7—figure supplement 1C*.

**Figure supplement 1—source data 4.** Original blot images for *Figure 7—figure supplement 1C*.

**Figure supplement 2.** SpAGS colocalizes with micromere-specific fate determinants.

control (Vasa-mCherry only, *Figure 10B*). This result suggests that SpAGS increases the frequency of micromere-like cell formation in pencil urchin embryos.

Sea urchin embryos show Vasa enrichment in micromeres at the 16-cell stage. In contrast, pencil urchin embryos show such enrichment later in the larval stage (3–4 dpf), which is more similar to the timing of the germline segregation in sea star embryos (*Juliano and Wessel, 2009*). We observed that SpAGS increased the Vasa signal enrichment in micromere-like cells compared to the control (Vasa-mCherry only) at the 16-cell stage. On the other hand, other AGS orthologs showed no significant difference from the control (*Figure 10C and D*). Nearly 80% (80.12% ± 3.75) of the SpAGS-injected embryos showed co-enrichment of AGS and Vasa in micromere-like cells, while the EtAGS and PmAGS groups showed only 49.2% ± 8.94 and 43.37% ± 3.94 enrichment, respectively (*Figure 10E*). Consistently, the SpAGS group showed the earlier segregation of Vasa-positive cells similar to sea urchin embryos at 1 dpf (*Figure 10F and G*), potentially accelerating the lineage segregation of the germline in the pencil urchin embryo.

## Discussion

The introduction of ACD in early embryogenesis of the sea urchin led to the formation of a new cell type, micromeres, with a critical organizer function. In the sea urchin, SpAGS is essential for micromere formation, while other echinoderm embryos show no cortical AGS localization (*Poon et al., 2019*). This study demonstrates that the GL1 motif of SpAGS is key for its vegetal cortical localization and function in micromere formation. Importantly, this unique role of the GL1 motif appears to be conserved across organisms. In *Drosophila* Pins and human LGN, GL1 is free from TPR binding, making it essential for the recruitment of Pins/LGN to the cortex (*Nipper et al., 2007*; *Takayanagi et al., 2019*). Thus, the evolutionary introduction of the GL1 motif into SpAGS likely increased recruitment affinity to the vegetal cortex, inducing ACD in the sea urchin embryo.

The GL1 deletion significantly disrupted micromere formation, while its replacement with other GL motifs had no effect. Therefore, the GL1 position rather than the sequence is essential for SpAGS function in ACD regulation. One exception to the above model is sea cucumber SbAGS, which has a putative GL1 motif yet does not induce ACD at the 16-cell stage. In this study, we found SbAGS is less localized to the vegetal cortex compared to SpAGS (*Figure 5—figure supplement 1*). Based on this observation, we speculate about three possible reasons why SbAGS is unable to facilitate ACD. First, the GL1 and GL2 of SbAGS are located too close to each other, compromising GL1's independence for recruitment. Indeed, the SpAGS-GL1GL2 mutant in which GL1 and GL2 are located next to each other showed compromised cortical localization and ACD function in the sea urchin embryo (*Figure 4G*). This suggests the distance between GL1 and GL2 might be critical for the GL1 to function properly. Second, a lack of GL4 in SbAGS loosens the autoinhibition state. The SpAGS mutant that lacks GL4 partially compromised ACD (*Figure 3F*), suggesting that the presence of GL4 is critical for ACD. Third, changes in the GL1 sequence of SbAGS compromise its recruiting efficacy. The results in *Figure 4* indicate that the position but not the sequence of GL1 is critical for ACD. However, we still cannot exclude the possibility that significant changes in the GL1 sequence of SbAGS compromised its function as a GL motif entirely. It will be critical to test all these possibilities directly in sea cucumber embryos in the future.

GL3 and GL4 sequences are also crucial for SpAGS activity, which appears to be conserved across organisms. In *Drosophila* Pins and human LGN, GL2-3 and GL3-4 sequences, respectively, are essential for their intramolecular interactions with TPR motifs, which control Pins/LGN's autoinhibited conformation (*Nipper et al., 2007*; *Pan et al., 2013*; *Smith and Prehoda, 2011*; *Takayanagi et al., 2019*). In a closed conformation, Pins/LGN are unable to bind to Mud/NuMA. Therefore, Gαi binding to GL1

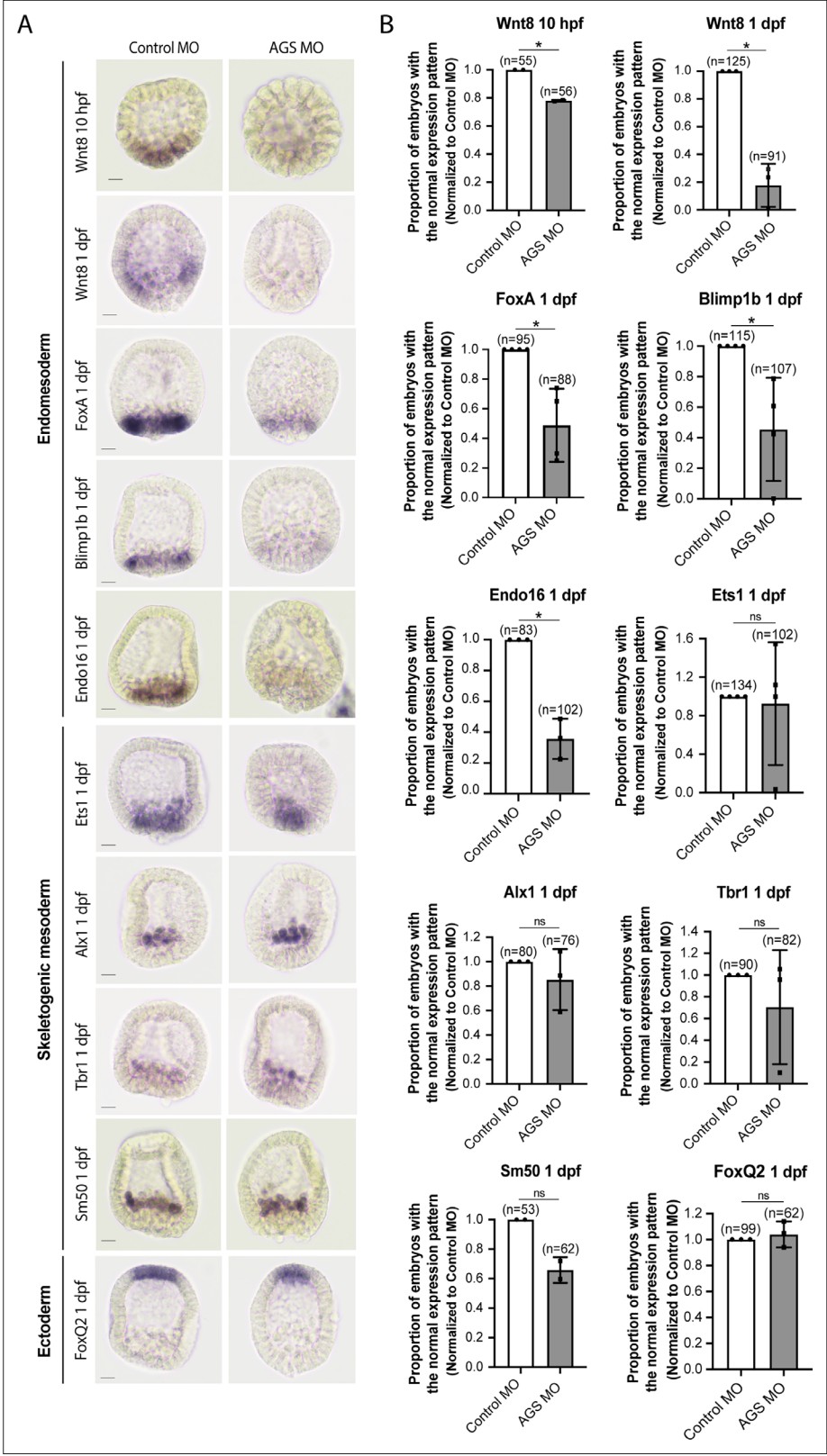

**Figure 8.** SpAGS is critical for the downstream gene expressions regulated by micromere signaling. Embryos were injected with 0.75 mM control MO or 0.75 mM SpAGS MO. Brightfield images (**A**) show the representative in situ hybridization (ISH) staining for each cell lineage marker scored in the corresponding graph (**B**). The number of embryos showing the normal signal patterns of each marker gene was scored and normalized to that of the

*Figure 8 continued on next page*

*Figure 8 continued*

control MO. Statistical analysis was performed by *t*-test. n indicates the total number of embryos scored. *p<0.05. Each experiment was performed at least two independent times. Error bars represent standard error. Scale bars = 20 µm.

The online version of this article includes the following source data and figure supplement(s) for figure 8:

**Source data 1.** Numerical data for *Figure 8B*.

**Figure supplement 1.** Sea urchin (SpDlg) and sea star (PmDlg) sequence alignment.

**Figure supplement 2.** Sea urchin (SpDlg) and sea star (PmDlg) sequence alignment.

relieves autoinhibition to activate them (*Du and Macara, 2004*; *Nipper et al., 2007*; *Takayanagi et al., 2019*; *Pan et al., 2013*). In this study, AGS1111 and AGS2222 mutants mostly showed uniform localization at every cortex in the embryo (*Figure 4F*). AGS2222 showed slightly more cortical localization than AGS1111, suggesting that GL2 may have a better affinity for TPR motifs; yet, it was still insufficient to induce micromere formation. Similarly, the mutants lacking the TPR4-6 motifs also showed randomized AGS localization (*Figure 2I*). These results suggest that the interactions between TPR and GL motifs are essential for restricting SpAGS localization to the vegetal cortex. Furthermore, these mutants all showed significant developmental toxicity, which was even worse than that of the SpAGS knockdown in some cases. Presumably, the constitutively open state of these SpAGS mutants interacts with other ACD factors in an uncontrolled manner, resulting in abnormal polarity induction in the embryo. Collectively, these results suggest that fine control of SpAGS close/open conformation is essential for accurate ACD regulation.

Another remaining question is how the SpAGS linker domain facilitates cortical localization but not function (*Figure 5F*). A previous study suggests that Aurora A phosphorylation at the Pins' linker domain partially controls the spindle orientation at the cortex (*Johnston et al., 2009*). It occurs independently of Gαi, and thus, it is proposed that Pins remains inactive in this process. Further, a recent study suggests that cortical localization of Pins is insufficient and requires Dlg and Insc to control spindle orientation (*Neville et al., 2023*). These studies suggest that Pins could be recruited through the linker domain to the cortex while remaining inactive. Therefore, we speculate the SpAGS-PmGL mutant that contains the SpLinker domain was recruited through this domain while remaining inactive in this study, which needs to be further tested in the future. In contrast, the PmAGS-SpLinker mutant that contains the restored Aurora A phosphorylation site in PmAGS did not recover the cortical localization (*Figure 5—figure supplement 2G*). Therefore, restoring the Aurora A site is insufficient to recruit PmAGS to the cortex, even in the sea urchin embryo. The SpAGS linker domain appears to have multiple putative phosphorylation sites besides the Aurora A site, which are absent in the PmAGS linker domain (*Figure 5—figure supplement 2A*). Therefore, it will be crucial to test in the future whether other phosphorylation sites of the linker domain contribute to the cortical recruitment of SpAGS independently of Gαi.

While the role of AGS protein in spindle orientation has been established in several model organisms, it was unknown if or how far AGS could regulate the fate determinants to facilitate ACD diversity. In this study, we learned that SpAGS is essential for the recruitment of ACD factors, such as Insc and NuMA, and fate determinants, such as Vasa and β-catenin, to micromeres. Notably, in pencil urchin embryos, SpAGS recruited Vasa protein into micromeres, suggesting SpAGS may be sufficient to recruit necessary fate determinants to create cell lineage segregation in another species. Although such lineage segregation of micromeres may be mediated solely by ACD, their function as organizers might require additional changes in the developmental program of the entire embryo. For example, sea urchin embryos have a robust hyaline layer to keep blastomeres together, which presumably increases the cell–cell interaction and may also enhance cell signaling during early embryogenesis. In contrast, a hyaline layer is not or little present in sea star or pencil urchin embryos, respectively. At present, we do not know what developmental changes are upstream or downstream of micromere formation during sea urchin diversification. It will be essential to identify in the future how far SpAGS impacts the developmental program other than inducing ACD and what other developmental elements play critical roles in establishing micromeres as a new cell lineage and organizers during sea urchin diversification.

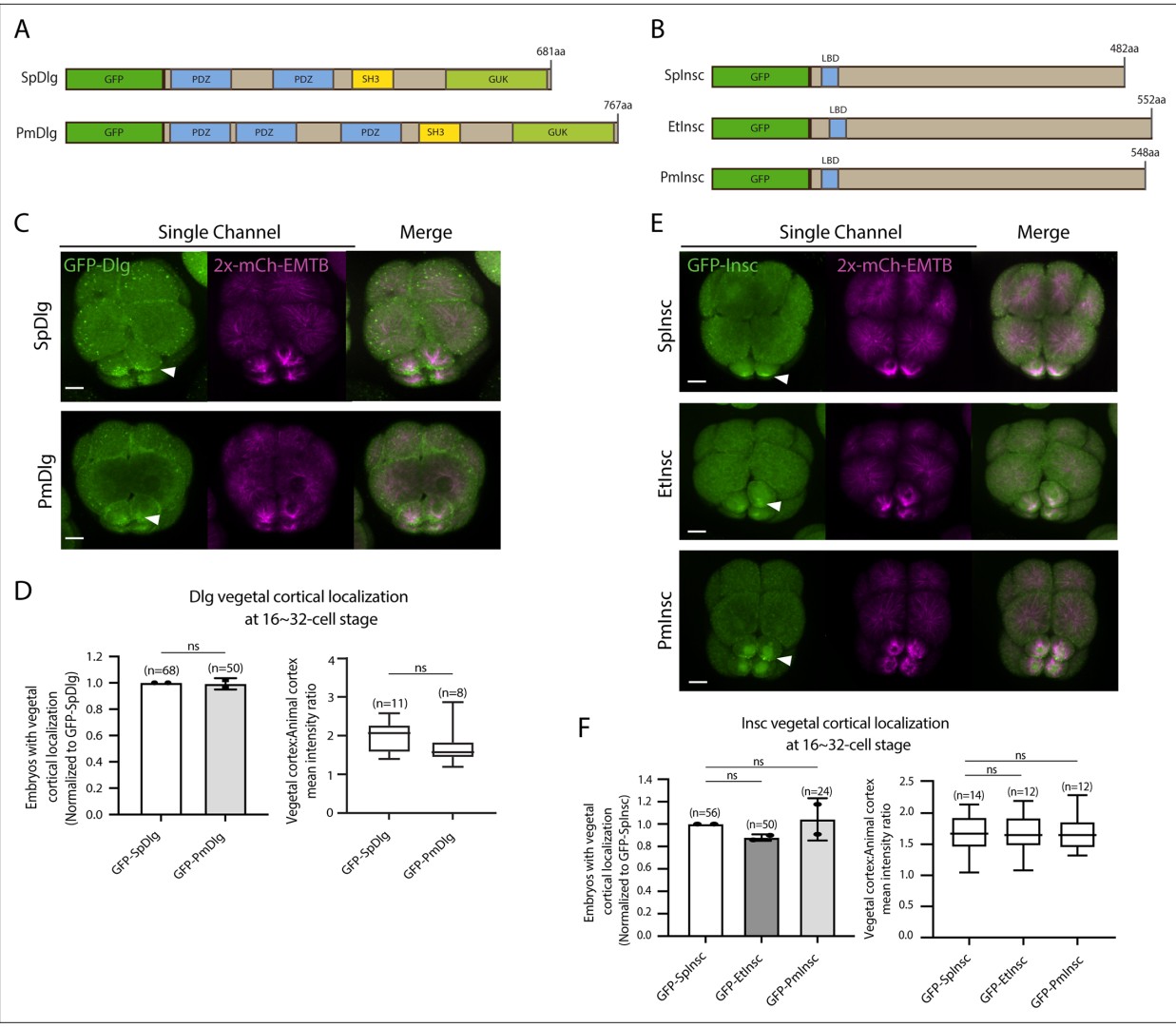

**Figure 9.** Dlg and Insc are not the variable factors of the asymmetric cell division (ACD) machinery in evolution. (**A, B**) Design of GFP-Dlg (**A**) and GFP-Insc (**B**) constructs that were tested in this study. Of note, EtDlg was unavailable in the database due to the limited genomic information available for this species. (**C–F**) Representative 2D-projection images of sea urchin embryos at the 8–16-cell stage showing localization of each echinoderm GFP-Dlg (**C**) and GFP-Insc (**E**). Z-stack images were taken at 1 µm intervals. Embryos were injected with 0.5 µg/µl stock of GFP-Dlg or GFP-Insc mRNAs and 0.25 µg/µl stock of 2x-mCherry-EMTB mRNA. White arrowheads indicate vegetal cortical localization of GFP constructs. The number of embryos with vegetal cortical localization of GFP-Dlg (**D**) and GFP-Insc (**F**) in 16–32-cell embryos was scored and normalized to that of the GFP-SpDlg or GFP-SpInsc (left graph). Right graph shows the ratio of the vegetal cortex-to-animal cortex mean intensity. Statistical analysis was performed by *t*-test or one-way ANOVA. n indicates the total number of embryos scored. Each experiment was performed at least two independent times. Error bars represent standard error. Scale bars = 10 µm.

The online version of this article includes the following source data for figure 9:

**Source data 1.** Numerical data for *Figure 9D*.

**Source data 2.** Numerical data for *Figure 9F*.

Overall, we conclude that the GL1 motif unique to sea urchin AGS orthologs is critical for SpAGS function in micromere formation. Since the unique role of the GL1 motif appears to be conserved across organisms, including *Drosophila* and humans, it is possible that the GL1 motif was once lost in the echinoderm common ancestor and recovered during sea urchin diversification. The recovery of this GL1 motif also resumed the interaction between SpAGS and other ACD machinery, such as NuMA, Insc, and Dlg, at the cortex in a similar manner to its orthologs Pins and LGN in other phyla, resulting in the controlled ACD and further interactions with fate determinants to form a new cell type in the sea urchin embryo. Therefore, unlike random unequal cell divisions that do not alter cell

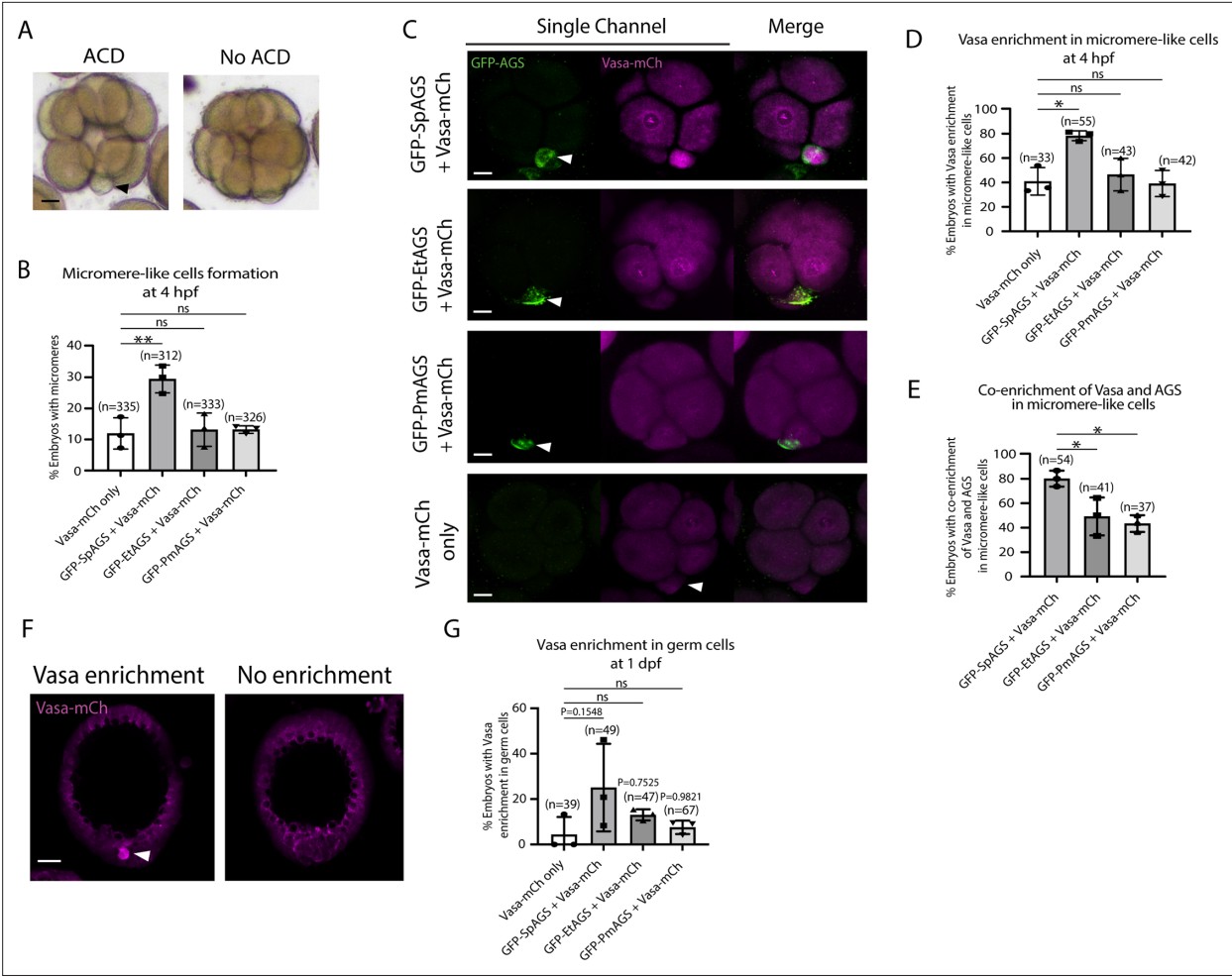

**Figure 10.** SpAGS but not EtAGS and PmAGS can induce a functional asymmetric cell division (ACD) in pencil urchin (Et) embryos. (**A**) Representative brightfield images of pencil urchin (Et) embryos with or without micromere-like cells. Black arrowhead indicates micromere-like cells. (**B**) Et embryos were injected with 0.3 μg/μl stock of GFP-AGS mRNAs and 1 μg/μl stock of Vasa-mCherry mRNA. The number of embryos making micromere-like cells was scored and normalized to that of the Vasa-mCherry only. Statistical analysis was performed against Vasa-mCherry only by one-way ANOVA. (**C**) Representative 2D-projection images of the injected Et embryos. Z-stack images were taken at 1 μm intervals to cover a layer of the embryo. White arrowheads indicate micromere-like cells. Scale bars = 10 μm. (**D**) The number of embryos showing Vasa enrichment in the micromere-like cells was scored and shown in percentage (%). Only the embryos that formed micromere-like cells were scored. Statistical analysis was performed against Vasa-mCherry only by one-way ANOVA. (**E**) Percentage of total embryos showing co-enrichment of Vasa and AGS in the micromere-like cells. Statistical analysis was performed against GFP-SpAGS by one-way ANOVA. (**F, G**) Representative 2D-projection images of Et embryos at 1 dpf. White arrowhead indicates Vasa enrichment in germ cells. Z-stack images were taken at 1 μm intervals. Percentage of total embryos showing Vasa enrichment in germ cells at 1 dpf was scored. Et embryos were injected with 0.3 μg/μl stock of GFP-AGS mRNAs. Statistical analysis was performed against Vasa-mCherry only by one-way ANOVA. n indicates the total number of embryos scored. *p<0.05 **p<0.01. Each experiment was performed at least three independent times. Error bars represent standard error. Scale bars = 20 μm.

The online version of this article includes the following source data for figure 10:

**Source data 1.** Numerical data for *Figure 10B*.

**Source data 2.** Numerical data for *Figure 10D*.

**Source data 3.** Numerical data for *Figure 10E*.

**Source data 4.** Numerical data for *Figure 10G*.

fates, AGS-mediated cell divisions appear to be highly organized and may be programmed to cause cell fate changes. Considering significant variations within the C-terminus of AGS orthologs and their immediate impact on micromere formation, we propose that AGS is a variable factor in facilitating ACD diversity among echinoderm embryos, contributing to developmental diversity within a phylum. Future studies in other taxa are awaited to demonstrate this concept further.

# Materials and methods

## Key resources table

| Reagent type (species) or resource | Designation | Source or reference | Identifiers | Additional information |
|---|---|---|---|---|
| Antibody | Anti-SpAGS (rabbit serum) | *Voronina and Wessel, 2006*; doi:10.1111/j.1440-169X.2006.00895.x | N/A | IF (1:300) PLA (1:300) |
| Antibody | Anti-Gαi (mouse monoclonal) | Santa Cruz Biotech | sc-56536 | IF (1:30) |
| Antibody | Anti-β-catenin (rabbit polyclonal) | *Yazaki et al., 2015*; doi:10.1017/S0967199414000033 | N/A | IF (1:300) |
| Antibody | Anti-SpInsc (rabbit polyclonal) | This article | N/A | See 'Insc antibody production and validation' WB (1:2000) IF (1:200) |
| Antibody | Anti-SpNuMA (rabbit polyclonal) | *Poon et al., 2019*; doi:10.1038/s41467-019-11560-8 | N/A | IF (1:500) |
| Antibody | Anti-β-actin (8H10D10) (mouse monoclonal) | Cell Signaling Technology | 3700S | WB (1:5000) |
| Antibody | Anti-Flag (mouse monoclonal) | MilliporeSigma | F1804 | PLA (1:100) |
| Antibody | Alexa 488-conjugated goat anti-rabbit IgG (goat polyclonal) | Cell Signaling Technology | 4412 | IF (1:300) |
| Antibody | Alexa 555-conjugated goat anti-mouse IgG (goat polyclonal) | Cell Signaling Technology | 4409 | IF (1:300) |
| Antibody | HRP-conjugated anti-Protein A antibody (goat polyclonal) | Abcam | ab7245 | WB (1:2000) |
| Antibody | HRP-conjugated goat anti-mouse IgG (horse polyclonal) | Cell Signaling Technology | 7076 | WB (1:2000) |
| Antibody | Anti-Digoxigenin-AP, Fab fragments (sheep polyclonal) | Roche | 11093274910 | ISH (0.1–0.5 ng/µl) |
| Chemical compound, drug | Hoechst 33342 | Thermo Fisher Scientific | 62249 | IF (1:1000) |
| Chemical compound, drug | Tris buffered saline, with tween (TBST) | MilliporeSigma | T9039 | |
| Chemical compound, drug | Tris-MOPS-SDS Running Buffer | GenScript | M00138 | |
| Chemical compound, drug | Transfer buffer powder | GenScript | M00139 | |
| Chemical compound, drug | DIG RNA labeling mix | Roche | 11277073910 | |
| Commercial assay or kit | mMESSAGE mMACHINE SP6 Transcription Kit | Thermo Fisher Scientific | AM1340 | |
| Commercial assay or kit | MEGAscript SP6 Transcription kit | Thermo Fisher Scientific | AM1330 | |
| Commercial assay or kit | MEGAscript T7 Transcription kit | Thermo Fisher Scientific | AM1333 | |
| Commercial assay or kit | In-Fusion HD Cloning | Clontech | 639648 | |
| Commercial assay or kit | Duolink In Situ Red Starter Kit Mouse/Rabbit | MilliporeSigma | DUO92101 | |

*Continued on next page*

*Continued*

| Reagent type (species) or resource | Designation | Source or reference | Identifiers | Additional information |
|---|---|---|---|---|
| Recombinant DNA reagent | Plasmid: SpAGS-GFP | *Poon et al., 2019*; doi:10.1038/s41467-019-11560-8 | N/A | |
| Recombinant DNA reagent | Plasmids: SpAGS-dC-term-GFP, SpAGS-dGL1/2/3/4-GFP, SpAGS1111/2222/2134/4234-GFP | This article | N/A | See 'Plasmid construction' |
| Recombinant DNA reagent | Plasmids: SpAGS-dN-term-GFP | This article | N/A | See 'Plasmid construction' |
| Recombinant DNA reagent | Plasmid: SpAGS-mCherry | *Poon et al., 2019*; doi:10.1038/s41467-019-11560-8 | N/A | |
| Recombinant DNA reagent | Plasmid: GFP-SpAGS, GFP-EtAGS, GFP-PmAGS, 2x-GFP-SpAGS, 2x-GFP-SbAGS | This article | N/A | See 'Plasmid construction' |
| Recombinant DNA reagent | Plasmid: GFP-SpAGS4444/GL1GL2/LGNGL/DmGL/EtGL/AGS3GL/PmGL/S389A/AGS3GL-3S/A/AGS3GL-GL2GL3 | This article | N/A | See 'Plasmid construction' |
| Recombinant DNA reagent | Plasmid: GFP-PmAGS-SpLinker | This article | N/A | See 'Plasmid construction' |
| Recombinant DNA reagent | Plasmid: GFP-SpDlg/PmDlg | This article | N/A | See 'Plasmid construction' |
| Recombinant DNA reagent | Plasmid: GFP-SpInsc/EtInsc/PmInsc | This article | N/A | See 'Plasmid construction' |
| Recombinant DNA reagent | Plasmid: GFP-NuMA | This article | N/A | See 'Plasmid construction' |
| Recombinant DNA reagent | Plasmid: mCherry-NuMA | This article | N/A | See 'Plasmid construction' |
| Recombinant DNA reagent | Plasmid: GFP-Par3 | This article | N/A | See 'Plasmid construction' |
| Recombinant DNA reagent | Plasmid: Vasa-GFP | *Yajima and Wessel, 2011*; doi:10.1242/dev.054940 | N/A | |
| Recombinant DNA reagent | Plasmid: Vasa-mCherry | *Uchida and Yajima, 2018*; doi:10.1016/j.ydbio.2018.06.015 | N/A | |
| Recombinant DNA reagent | Plasmid: 3xFlag-GFP-SpAGS/SpDlg/SpNuMA, 3xFlag-Vasa-GFP | This article | N/A | See 'Plasmid construction' |
| Recombinant DNA reagent | Plasmid: 2x-mCherry-EMTB | Addgene | 26742 | |
| Software, algorithm | EchinoBase | http://www.echinobase.org/Echinobase/ | N/A | Echinoderm protein sequences |
| Software, algorithm | NCBI blast | https://blast.ncbi.nlm.nih.gov/Blast.cgi | N/A | Protein motif search |
| Software, algorithm | Clustal Omega | https://www.ebi.ac.uk/Tools/msa/clustalo/ | N/A | Protein sequence alignment |
| Software, algorithm | ImageJ | https://imagej.nih.gov/ij/ | N/A | Quantitative analysis |
| Software, algorithm | GraphPad PRISM 8 | https://www.graphpad.com/scientific-software/prism/ | N/A | Statistical analysis |
| Sequence-based reagent | SpAGS-MO | NM_001040405.1 | Morpholino antisense oligos | GGCCCGTTTCACAAAGCCTTTGTTT |

*Continued on next page*

*Continued*

| Reagent type (species) or resource | Designation | Source or reference | Identifiers | Additional information |
|---|---|---|---|---|
| Sequence-based reagent | SpAlx1 | XM_011663478.2 | ISH probe primers | F: GGATATTTTCTCGACCGGGA TC R: CGAGTAACCGTTCATCATCC CC |
| Sequence-based reagent | SpBlimp1b | NM_214574.3 | ISH probe primers | F: ATGGGGTGCAACGACAACGC CGTG R: CTATGATTTGTTCGTACGAT TGAG |
| Sequence-based reagent | SpEndo16 | NM_214519.1 | ISH probe primers | F: GCAGAGTTCAACAGAA TCGAC R: GCCAGTAGACGTAGCAGAAG |
| Sequence-based reagent | SpEts1 | XM_030976919.1 | ISH probe primers | F: TCAATCATGGCGTCTATGCA CTG R: ACAGCTGCAGGGATAACAGG |
| Sequence-based reagent | SpFoxA | NM_001079542.1 | ISH probe primers | F: ATGGCCAATAGTGCCATGAT CTCG R: TCACATTGCATGGTTT GCTTG |
| Sequence-based reagent | SpFoxQ2 | XM_003731512.3 | ISH probe primers | F: ATGACTTTATTCAGCATTGA CAAC R: TAGCAGGATCCTACAGAAGA CCAG |
| Sequence-based reagent | SpSm50 | NM_214610.3 | ISH probe primers | F: ATGAAGGGAGTTTTGTTTAT TGTGG CTAGTC R: GTTATGCCAACGCGTCTGCC TCTTG AAGC |
| Sequence-based reagent | SpTbr1 | XM_786173.5 | ISH probe primers | F: CCACCGCTGCACCAGACGAC R: CTGCCGGCTGGCGCCAATTG CG |
| Sequence-based reagent | SpWnt8 | NM_214667.1 | ISH probe primers | F: ATGGATGTTTTTACGGAATT TGTTCG R: CTACAGCCTCGATCCAACGG GCTG |

## Animals and echinoderm embryos

*S. purpuratus* (sea urchins) were collected from the ocean by Pat Leahy, Kerchoff Marine Laboratories, California Institute of Technology, or Josh Ross, South Coast Bio-Marine LLC, Long Beach, CA, and kept in an aquarium cooled to 16°C. *E. tribuloides* (pencil urchins) were collected from the ocean by KP Aquatics LLC., Tavernier, FL, and maintained in the aquarium at room temperature. Gametes were acquired via 0.5 M KCl injections. Eggs were collected in seawater (SW), and sperm was collected dry. For injection, eggs were de-jellied using pH 4.0 SW and placed in a plate coated with protamine sulfate. These eggs were then fertilized and injected in the presence of 1 mM 3-amino triazole (Sigma, St. Louis, MO) to prevent crosslinking of fertilization envelopes, and embryos were cultured in SW at 16°C. For protein collection for immunoprecipitation, eggs were fertilized in 1 mM 3-amino triazole. Fertilization envelopes were removed by pipetting, and fertilized eggs were placed in a plate coated with the fetal bovine serum to prevent eggs from sticking to the plate.

## Plasmid construction

All constructs were prepared in pSP64 or pCS2 vectors, which were optimized for in vitro transcription. SpAGS was previously identified in the sea urchin (*Voronina and Wessel, 2006*) and SpAGS-GFP was constructed by PCR amplification of the SpAGS ORF, then subcloned into the pSp6 β-globin UTR plasmid between the *Xenopus* β-globin 5′ and 3′ UTRs as described in *Poon et al., 2019* (*Figure 2—figure supplement 1A*). To remove GL1 (473aa DNFFEALSRFQSNRMDEQRCSF 495aa) from SpAGS-GFP, the internal Bbvc1 (458a) and Bsm1 (532aa) sites were used to remove the sequence, including GL1, and the corresponding sequence lacking only GL1 (gBlock, IDT, IA) was fused back using In-Fusion HD Cloning kit according to manufacturer's protocol (#639648, Clontech, USA) (*Figure 2—figure*

*supplement 1B*). The other C-terminal deletion constructs were created following the same method using the internal BbvC1 (458aa) and vector Apa1 sites to remove the original sequence and replace it with each DNA fragment (gBlock, IDT) with the desired sequence. The N-terminal deletion constructs were constructed by removing the entire AGS ORF from the SpAGS-GFP plasmid using the vector Bgl2 and Apa1 sites, then replacing it with a custom DNA fragment (gBlock, IDT), each with the appropriate deletion. The ORF of SbAGS (GenBank ID is GAUT01023097.1) was synthesized by IDT, which was then inserted into the Sp64-2xGFP vector at Not1 and Spe1 sites. The ORF of Insc, Dlg, NuMA, and Par3 was PCR amplified and subcloned into the pSP64-GFP/mCherry vector. The 3xFlag DNA fragment (gBlock, IDT) was inserted into pSP64-GFP-SpInsc/SpDlg/NuMA and pSP64-Vasa-GFP for PLA analysis. pCS2-2x-mCherry-EMTB (#26742 Addgene) (*Miller and Bement, 2009*) was obtained from Addgene. pSP64-Vasa-mCherry was previously constructed in *Uchida and Yajima, 2018*. pSP64-Vasa-GFP and pSP64-AGS-mCherrywere previously built and used (*Fernandez-Nicolas et al., 2022*; *Poon et al., 2019*; *Yajima and Wessel, 2011*; *Yajima and Wessel, 2015*).

## mRNA injection and microscopy

Constructs were linearized with the appropriate restriction enzymes overnight (Not1 for pCS2-2x-mCherry-EMTB constructs, Smal, Sall, or EcoRI for all pSP64 constructs), then transcribed in vitro with mMESSAGE mMACHINE SP6 Transcription Kit (#AM1340, Thermo Fisher Scientific), which involved a 4 hr incubation at 37°C, followed by a DNaseI treatment and LiCl precipitation overnight at –20°C. Sea urchin embryos were injected at the one-cell stage with 0.15–1 µg/µl of each mRNA as individually indicated. A morpholino antisense oligonucleotide (MO) that explicitly blocks the translation of SpAGS was previously designed and used in *Poon et al., 2019*. The SpAGS MO sequence is listed below (Table S1Key resources table). For knockdown experiments, embryos were co-injected with 0.75 mM MO with or without 0.15 µg/µl of SpAGS-GFP mRNA. Embryos were imaged using the Nikon CSU-W1 Spinning disk laser microscope.

## Insc antibody production and validation

Three affinity-purified rabbit antibodies against SpInsc were made by GenScript (Piscataway, NJ). Antibody #1 showed the most specific vegetal cortex signal by immunofluorescence (*Figure 7—figure supplement 1A*). This antibody detected multiple bands yet still displayed the primary band at the expected size (53 kDa) by immunoblot (*Figure 7—figure supplement 1B*). The competition assay with SpInsc-peptide removed all bands except for the band at 15 kDa (*Figure 7—figure supplement 1C*). Thus, the larger bands detected by this antibody may be the complexes of Insc proteins since Insc is known to form dimers and hexamers with LGN (*Culurgioni et al., 2018*).

## Immunoblotting

Samples were run on a 10% Tris-glycine polyacrylamide gel (Invitrogen, Carlsbad, CA) before transfer on a nitrocellulose membrane for immunoblotting with Insc antibodies used at 1:2000 dilution with 1.5% BSA, or Actin (#3700S, Cell Signaling Technology) antibody at 1:5000 dilution with 0% BSA, followed by treatment with HRP-conjugated anti-Protein A (ab7245, Abcam) for Insc or HRP-conjugated anti-mouse (#7076, Cell Signaling Technology) secondary antibody for Actin at 1:2000. The reacted proteins were detected by incubating the membranes in the chemiluminescence solution (luminol, coumaric acid, hydrogen peroxide, Tris pH 6.8) and imaged by the ChemiDoc Gel Imaging System (Bio-Rad, USA).

## Immunofluorescence

The final concentrations of primary antibodies were anti-SpInsc at 1:200, anti-SpAGS (*Poon et al., 2019*) and anti-β-catenin (*Yazaki et al., 2015*) at 1:300, anti-SpNuMA (*Poon et al., 2019*) at 1:500, and anti-Gαi (#sc-56536, Santa Cruz Biotech) at 1:30. The secondary antibodies were used at a dilution of 1:300 Alexa 488-conjugated goat anti-rabbit (#4412, Cell Signaling Technology) or Alexa 555-conjugated goat anti-mouse (#4409, Cell Signaling Technology). Hoechst dye (#62249, Thermo Fisher Scientific) at 1:1000 (10 mg/ml stock) was used to visualize DNA. Embryos of the desired developmental stage were fixed with 90% cold methanol for more than 1 hr at –20°C, washed with 1× PBS, and incubated with the primary antibody overnight at 4°C, followed by 10 washes with 1× PBS, then incubated with the secondary antibody at room temperature for 3 hr. The secondary antibody was

washed 10 times with 1× PBS and Hoechst treatment for 15 min. Samples were plated onto slides. All fluorescent images were taken under the Nikon CSU-W1 Spinning disk laser microscope.

## Proximity ligation assay

Embryos at the 8–16-cell stage were fixed with 90% cold methanol for over 1 hr at –20°C, washed with 1× PBS, and treated with 0.05% Triton-X for 15 min. PLA was processed following a manufacturer's protocol (#DUO92101, MilliporeSigma). The concentration of primary antibodies was anti-SpAGS (*Voronina and Wessel, 2006*) at 1:300 and anti-Flag (#F1804, MilliporeSigma) at 1:100. Embryos were taken images under the Nikon CSU-W1 Spinning disk laser microscope.

## In situ hybridization

The embryos were fixed using 4% paraformaldehyde at the ideal stage. Fixed embryos were washed with MOPS buffer and stored in 70% EtOH at –20°C until needed. ISH was performed as previously described (*Minokawa et al., 2004*; *Perillo et al., 2021*). Sequences used to make antisense probes were PCR amplified from 1 dpf embryonic cDNA of sea urchin using the primers listed in the literature and Key resources table (*Rizzo et al., 2006*; *Ettensohn et al., 2003*; *Cary et al., 2017*) and cloned into TOPO vector (#45-124-5, Thermo Fisher Scientific) (Key resources table). The TOPO plasmids were linearized using BamHI or HindIII (T7 transcription) and NotI or XhoI (SP6 transcription) for subsequent in vitro transcription using either SP6 or T7 MEGAscript Transcription kit (#AM1330 or AM1333, Thermo Fisher Scientific) with DIG RNA labeling mix (#11277073910, Roche; Indianapolis, IN).

## Data analysis

All quantitative data were analyzed using GraphPad Prism 8.3.1 software. Each experiment was repeated at least two independent times. Statistical significance was determined by a *t*-test or one-way ANOVA.*p<0.05, **p<0.01, ***p<0.001, and ****p<0.0001.

## Blast and motif analysis

All echinoderm sequences were obtained from Echinobase.org. Protein sequence alignment and molecular phylogenetic tree were constructed using *Clustal Omega* and *CIPRES Science Gateway V. 3.3*. Protein structural motif analysis was performed through the NCBI blast search of the database CDD v3.17 with the value threshold of 0.02. The GoLoco (GL) motif found in the C-terminal of AGS-family proteins is defined by a conserved core of 19 amino acids except for the *C. elegans*, where the single GL motif is 18 amino acids long (*Willard et al., 2004*). In *Figure 1B*, some GL or TPR motifs were considered partial as they are predicted to be less than 18 amino acids long, or a few amino acids are altered in the motif, respectively. Each GL motif was numbered according to sequence similarity to that of *S. purpuratus* AGS GL motifs.

## Materials availability statement

Major plasmid constructs made in this study are available through Addgene upon completion of the depository process. All other materials are available through the corresponding author upon reasonable requests.

# Acknowledgements

We would like to thank Mr. Ronit Sethi for providing assistance in identifying the optimal conditions for AGS-MO and OE experiments. NE and FDMW were responsible for the concept, experimental design and undertaking, data analysis, and manuscript construction and editing regarding all bioinformatics analyses; AF was responsible for initial conceptualization, experimental design, undertaking, and data analysis; MY was responsible for concepts, experimental design and undertaking, data analysis, manuscript construction, and editing for all sections.This work was supported by NSF (IOS-1940975) and NIH (1R01GM126043-01).

## Additional information

### Funding

| Funder | Grant reference number | Author |
|---|---|---|
| National Science Foundation | IOS-1940975 | Mamiko Yajima |
| National Institute of General Medical Sciences | 1R01GM126043-01 | Mamiko Yajima |

The funders had no role in study design, data collection and interpretation, or the decision to submit the work for publication.

### Author contributions

Natsuko Emura, Florence DM Wavreil, Conceptualization, Data curation, Software, Formal analysis, Validation, Investigation, Visualization, Methodology, Writing – original draft, Writing – review and editing; Annaliese Fries, Data curation, Investigation, Visualization; Mamiko Yajima, Conceptualization, Supervision, Funding acquisition, Investigation, Methodology, Writing – original draft, Project administration, Writing – review and editing

### Author ORCIDs

Natsuko Emura (iD) http://orcid.org/0000-0003-0861-6048
Florence DM Wavreil (iD) http://orcid.org/0000-0003-4316-9669
Mamiko Yajima (iD) https://orcid.org/0000-0002-5613-1005

Reviewer #1 (Public review): https://doi.org/10.7554/eLife.100086.3.sa1
Reviewer #2 (Public review): https://doi.org/10.7554/eLife.100086.3.sa2
Author response https://doi.org/10.7554/eLife.100086.3.sa3

## Additional files

### Supplementary files

• MDAR checklist

### Data availability

All data generated or analyzed during this study are included in the manuscript and supporting files; source data contains the numerical data used to create graphs in the figures.

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
