## [Editor Report · eLife Assessment]

This **important** study presents work on the molecular mechanism driving asymmetric cell division and fate decisions during embryonic development of echinoids. The evidence supporting the claims of the authors is **convincing**. The work will be of interest to developmental biologists and cell biologists working in the field of self-renewal.

---

## [Referee Report · Reviewer #1 (Public review)]

Summary:

Previous work has shown that the evolutionarily-conserved division-orienting protein LGN/ Pins/ GPR1/2 (vertebrates/flies/nematodes) participates in division orientation across a variety of cell types, perhaps most importantly those that undergo asymmetric divisions (ACDs). Micromere formation in echinoids relies on asymmetric cell division at the 16-cell stage, and these authors previously demonstrated a role for the LGN/Pins homolog AGS (Activator of G-protein signaling) in that ACD process. Here they extend that work by investigating and exploiting the question of why echinoids but not other echinoderms form micromeres. Using an impressive combination of phylogenetics and molecular experiments, they determine that much of the difference in ACD and micromere formation in echinoids can be attributed to differences in the AGS C-terminus, in particular a GoLoco domain (GL1) that is missing in most other echinoderms. This work helps explain how AGS works and thereby enhances our understanding of a conserved player in division orientation.

---

## [Referee Report · Reviewer #2 (Public review)]

This study from Dr. Emura and colleagues addresses the relevance of AGS3 mutations in the execution of asymmetric cell divisions promoting the formation of the micromere during sea-searching development. To this aim, the authors use quantitative imaging approaches to evaluate the localisation of AGS3 mutants truncated at the N-terminal region or at the C-terminal region, and correlate these distributions with the formation of micromere and correct development of embryos to the pluteus stage. The authors also analyse the capacity of these mutated proteins to rescue developmental defects observed upon AGS3 depletion by morpholino antisense nucleotides (MO). Collectively these experiments revealed that the C-terminus of AGS3, coding for four GoLoco motifs binding to cortical Gaphai proteins, is the molecular determinant for cortical localisation of AGS3 at the micromeres and correct pluteus development. Further genetic dissections and expression of chimeric AGS3 mutants carrying shuffled copies of the GoLoco motifs or four copies of the same motifs revealed that the position of GoLoco1 is essential for AGS3 functioning. To understand whether the AGS3-GoLoco1 evolved specifically to promote asymmetric cell divisions, the author analyse chimeric AGS3 variants in which they replaced the sea urchin GoLoco region with orthologs from other echinoids that do not form micromeres, or from *Drosophila* Pins or human LGN. These analyses corroborate the notion that the GoLoco1 position is crucial for asymmetric AGS3 functions. In the last part of the manuscript, the authors explore whether SpAGS3 interacts with the molecular machinery described to promote asymmetric cell division in eukaryotes, including Insc, NuMA, Par3 and Galphai, and show that all these proteins colocalize at the nascent micromere, together with the fate determinant Vasa. Collectively this evidence highlighted how evolutionarily selected AGS3 modifications are essential to sustain asymmetric divisions and specific developmental programs associated with them.

The manuscript addresses an interesting question and uses elegant genetic approaches associated with imaging analyses to elucidate the molecular mechanisms whereby AGS3 and spindle orientation proteins promote asymmetric divisions and specific developmental programs. This considered, it might be worth clarifying a few aspects of the reported findings.

(1) In some experimental settings, the presence of AGS3 mutants exacerbates the AGS3 deletion by MO (Fig. 4F). Can the author speculate on what can be the molecular explanation?

(2) Imaging analyses of Figure 4B-C suggest that the mutant AGS1111 does not localise at the vegetal cortex while AGS2222 does (Fig. 4C). However these mutants induce similar developmental defects (Fig. 4F) . What could be the reason?

(3) Figure 7 shows the crosstalk between AGS3 and other asymmetry players including NuMA. Vertebrate and *Drosophila* NuMA are ubiquitously present in tissues and localises to the spindle poles in mitosi. However in Figure 7A and 7E NuMA seems expressed only in a subset of sea urchin embryonic cells. Is this the case?

---

## [Author Response]

The following is the authors’ response to the original reviews.

**Public Reviews:**

**Reviewer #1 (Public Review):**
Summary:Previous work has shown that the evolutionarily-conserved division-orienting protein LGN/Pins (vertebrates/flies) participates in division orientation across a variety of cell types, perhaps most importantly those that undergo asymmetric divisions. Micromere formation in echinoids relies on asymmetric cell division at the 16-cell stage, and these authors previously demonstrated a role for the LGN/Pins homolog AGS in that ACD process. Here they extend that work by investigating and exploiting the question of why echinoids but not other echinoderms form micromeres. Starting with a phylogenetics approach, they determine that much of the difference in ACD and micromere formation in echinoids can be attributed to differences in the AGS Cterminus, in particular a GoLoco domain (GL1) that is missing in most other echinoderms.

Thank you for the summary.

Strengths:There is a lot to like about this paper. It represents a superlative match of the problem with the model system and the findings it reports are a valuable addition to the literature. It is also an impressively thorough study; the authors should be commended for using a combination of experimental approaches (and consequently generating a mountain of data).

Thank you.

Weaknesses:There is an intriguing finding described in Figure 1. AGS in sea cucumbers looks identical to AGS in the pencil urchin, at least at the C terminus (including the GL1 domain). Nevertheless, there are no micromeres in sea cucumbers. Therefore another mechanism besides GL motif organization has arisen to support micromere formation. It is a consequential finding and an important consideration in interpreting the data, but I could not find any mention of it in the text. That is a missed opportunity and should be remedied, ideally not only through discussion but also experimentation. Specifically: does sea cucumber AGS (SbAGS) ever localize to the vegetal cortex in sea cucumbers? Can it do so in echinoids? Will that support micromere formation?

Thank you for pointing this out.

To respond to the Reviewer’s request, we synthesized sea cucumber (Sb) AGS based on the sequence available in the database and tested it in the sea urchin (Sp) embryos, which is enclosed in Fig. S3. We performed this experiment to confirm that SbAGS localizes less at the vegetal cortex than SpAGS as a proof of principle. However, we hesitate to conduct further studies using the synthetic sequence in this study. Sea cucumbers are an emerging yet understudied model. This species is not readily available or established as a model system for embryology. Even for the two species (A. japonicus in Japan and P. parvimensis in the USA) that were previously used for embryonic studies, their gametes are typically available only for 12 months in a year. Since some echinoderm researchers are aiming to establish sea cucumbers as a model system in the near future (see 2024 review: PMID: 38368336), we hope to be able to have better access to their embryos in the future. Yet, it may require a few more years to reach that condition.

In this revised manuscript, we explained the above details and further added the discussion described below. All of the experimental models used in this study are wild animals obtained from the ocean, raising the standard for reproducibility. However, handling wild animals could come with challenges. We hope that the reviewer understands the unique benefits and challenges of this study.

Discussion:

Previous studies (PMIDs: 17726110; 21855794) suggest that GL1 is not involved in intramolecular interaction with TPR domains. This allows GL1 to interact independently with Gαi for cortical recruitment yet without influencing other GLs for AGS activation. To ensure GL1's independence, GL1 is typically located distantly from other GLs in Pins (flies), LGN (humans), and AGS (sea urchins). Based on this prior knowledge, we speculate three scenarios for sea cucumber (Sb) AGS not being able to localize or function during asymmetric cell division (ACD): (1) GL1 and GL2 are located too close to each other, compromising GL1's independence for recruitment. (2) A lack of GL4 loosens the autoinhibition state. (3) The GL1 sequence of SbAGS is quite different from that of echinoids’ AGS (Figure S2), compromising its recruiting efficacy.

For (1), we tested this possibility by making the SpAGS-GL1GL2 mutant that has GL1 and GL2 next to each other (Fig. 4G). This mutant indeed compromised its cortical localization and function in ACD. For (2), we showed that the lack of GL4 partially compromised ACD in SpAGS (Fig. 3F), suggesting that GL4 supports ACD. For (3), The results in Figure 4 indicate that the position but not the sequence of GL1 is critical for ACD. Based on these observations, we speculate a combination of (1) and (2) compromised SbAGS's ACD function. However, it is still possible that a significant difference in the GL1 sequence diminished its function as GL entirely. Future studies should address these remaining questions directly in the sea cucumber embryos once they are established as a model system in the near future (PMID: 38368336)

The authors point out that AGS-PmGL demonstrates enrichment at the vegetal cortex (arrow in 5G, quantifications in 5H), unlike PmAGS. AGS-PmGL does not however support ACD. They interpret this result to indicate "that other elements of SpAGS outside of its C-terminus can drive its vegetal cortical localization but not function." This is a critical finding and deserves more attention. Put succinctly: Vegetal cortical localization of AGS is insufficient to promote ACD, even in echinoids. Why should this be?

Thank you for the suggestion. We revised our wording to be more succinct. Of note, as we noted in the text, AGS-PmGL has only two GL domains, which will likely not provide the full force to control ACD and result in insufficient ACD function.

The authors did perform experiments to address this problem, hypothesizing that the difference might be explained by the linker region, which includes a conserved phosphorylation site that mediates binding to Dlg. They write "To test if this serine is essential for SpAGS localization, we mutated it to alanine (AGS-S389A in Fig. S3A). Compared to the Full AGS control, the mutant AGS-S389A showed reduced vegetal cortical localization (Fig. S3B-C) and function (Fig. S3D-E). Furthermore, we replaced the linker region of PmAGS with that of SpAGS (PmAGSSpLinker in Fig. S4A-B). However, this mutant did not show any cortical localization nor proper function in ACD (Fig. S4C-F). Therefore, the SpAGS C-terminus is the primary element that drives ACD, while the linker region serves as the secondary element to help cortical localization of AGS."The experiments performed only make sense if the AGS-PmGL chimeric protein used in Figure 5 starts the PmGL sequence only after the Sp linker, or at least after the Sp phosphorylation site. I can't tell from the paper (Figure S3 indicates that it does, whereas S5 suggests otherwise), but it's a critical piece of information for the argument.

Thank you for the pointer, and we apologize for the confusion. AGS-PmGL contains the SpAGS linker domain. To clarify this point, we added the amino acid position at the junction of each chimeric construct diagram in Figs. 5 and S4. To clarify, Figure S5 is about the GL domain mutations (not about the Linker).

Another piece of missing information is whether the PmAGS can be phosphorylated at its own conserved phosphorylation site. The authors don't test this, which they could at least try using a phosphosite prediction algorithm, but they do show that the candidate phosphorylation site has a slightly different sequence in Pm than in Et and Sp (Fig. S4A). With impressive rigor, the authors go on to mutate the PmAGS phosphorylation site to make it identical to Sp. Nothing happens. Vegetal cortical localization does not increase over AGS-PmGL alone. Micromere formation is unrescued.There is therefore a logic problem in the text, or at least in the way the text is written. The paragraph begins "Additionally, AGS-PmGL unexpectedly showed cortical localization (Figure 5G), while PmAGS showed no cortical localization (Figure 5B)." We want to understand why this is true, but the explanation provided in the remainder of the paragraph doesn't match the question: according to quite a bit of their own data, the phosphorylation site in the linker does not explain the difference. It might explain why AGS-PmGL fails to promote micromere formation, but only if the AGS-PmGL chimeric protein uses the Pm linker domain (see above).

Thank you for the insightful suggestion. As suggested, we performed the phosphosite predictions using GPS 6.0 (PMID**:** 37158278) and enclosed the results in Fig. S4A (replacing the old Fig. S3A). The software predicts SpAGS and EtAGS have a predicted AuroraA phosphorylation site (RRRSMEN in Supplemental figure S4A) in their linker domain, while PmAGS does not. Sp and Et AGS also have the additional 5-7 predicted phosphorylation sites, while PmAGS has only three sites with low scores. Therefore, the linker domain is not conserved in PmAGS.

The PmAGS+SpLinker mutant does restore the predicted AuroraA phosphorylation site on the software, yet it does not restore the cortical localization or ACD function in the embryo. Therefore, other sites in the Linker region might also be necessary for cortical localization and ACD function of AGS. In this study, we did not perform further manipulations in the Linker domain. As the reviewer rightfully pointed out, even if we identify the Linker regions essential for AGS localization and function, it will be difficult to interpret the result unless we know what proteins interact with the Linker domain of AGS. Therefore, this is beyond the scope of the current manuscript. We discussed these remaining matters in the discussion section.

Another concern that is potentially related is the measurement of cortical signal. For example, in the control panel of Figure 5C, there is certainly a substantial amount of "non-cortical" signal that I believe is nuclear. I did not see a discussion of this signal or its implications. My impression of the pictures generally is that the nuclear signal and cortical signal are inversely correlated, which makes sense if they are derived from the same pool of total protein at different points of the cell cycle. If that's the case (and it might not be) I would expect some quantifications to be impacted. For example, the authors show in Figure S3B that AGS-S389A mutant does not localize to the cortex. However, this mutant shows a radically different localization pattern to the accompanying control picture (AGS), namely strong enrichment in what I assume to be the nucleus. Is the S389 mutant preventing AGS from making it to the cortex? Or are these pictures instead temporally distinct, meaning that AGS hasn't yet made it out of the nucleus? Notably, the work of Johnston et al. (Cell 2009), cited in the text, does not show or claim that the linker domain impacts Pins localization. Their model is rather that Pins is anchored at the cortex by Gαi, not Dlg, and that is the same model described in this manuscript.In agreement with that model and the results of Johnston et al., a later study (Neville et al. EMBO Reports 2023) failed to find a role for Dlg or the conserved phosphorylation site in Pins localization.

In the sea urchin embryo, the dye or GFP often appears in the nucleus randomly on top of the cytoplasm (for example, see Fig. S2b of PMID: 35444184). Further, embryos tend to incorporate exogenous genomic fragments more efficiently during early embryogenesis (PMID: 3165895). It is proposed that early embryos may have a loosened or incomplete nuclear envelope compared to adult cells as they divide rapidly (every 40 minutes). Therefore, any excess protein with no specific localization signal may randomly appear in the nucleus as it serves as an available space in the cell. As the Reviewer rightfully pointed out, we consider that the nuclear AGS signal is due to the lack of a specific destination since this signal pattern is not consistent across embryos. In contrast, the proteins that have nuclear localization (e.g., transcription factors) usually show a consistent nuclear signal across cells and embryos with less cytoplasmic signal. To avoid confusion, we replaced the S389A image in Fig. S3B (which is now Fig. S4C) as well as any other images that may create similar confusion.

**Reviewer #2 (Public Review):**
This study from Dr. Emura and colleagues addresses the relevance of AGS3 mutations in the execution of asymmetric cell divisions promoting the formation of the micromere during seasearching development. To this aim, the authors use quantitative imaging approaches to evaluate the localisation of AGS3 mutants truncated at the N-terminal region or at the Cterminal region, and correlate these distributions with the formation of micromere and correct development of embryos to the pluteus stage. The authors also analyse the capacity of these mutated proteins to rescue developmental defects observed upon AGS3 depletion by morpholino antisense nucleotides (MO). Collectively these experiments revealed that the Cterminus of AGS3, coding for four GoLoco motifs binding to cortical Gaphai proteins, is the molecular determinant for cortical localisation of AGS3 at the micromeres and correct pluteus development. Further genetic dissections and expression of chimeric AGS3 mutants carrying shuffled copies of the GoLoco motifs or four copies of the same motifs revealed that the position of GoLoco1 is essential for AGS3 functioning. To understand whether the AGS3-GoLoco1 evolved specifically to promote asymmetric cell divisions, the authors analyse chimeric AGS3 variants in which they replaced the sea urchin GoLoco region with orthologs from other echinoids that do not form micromeres, or from *Drosophila* Pins or human LGN. These analyses corroborate the notion that the GoLoco1 position is crucial for asymmetric AGS3 functions. In the last part of the manuscript, the authors explore whether SpAGS3 interacts with the molecular machinery described to promote asymmetric cell division in eukaryotes, including Insc, NuMA, Par3, and Galphai, and show that all these proteins colocalize at the nascent micromere, together with the fate determinant Vasa. Collectively this evidence highlighted how evolutionarily selected AGS3 modifications are essential to sustain asymmetric divisions and specific developmental programs associated with them.

Thank you for the useful summary.

**Recommendations for the authors:**

**Reviewer #1 (Recommendations For The Authors):**
The quantifications of "vegetal cortical localization" are somewhat incomplete. As measured, "vegetal cortical localization" does not demonstrate particular enrichment at the vegetal cortex, only that some signal appears there. In other words, we can't tell for sure that there is any more signal at the vegetal cortex than anywhere else along the cortex, and in fact that's plainly true and even described for the ACS1111 and AGS2222 constructs. One solution would be to measure signal strength around the cell perimeter and see where it is strongest.

As suggested by the Reviewer, we added new measurements, focusing and comparing the signals on the animal versus vegetal cortices (Figs. 2C, 3D, 4C, 5C, &H, 9D & F, S3D, S4D &I).

A related issue is that the strength of cortical enrichment is indicated in this paper by the ratio of cortical to "non-cortical" signal, but "non-cortical" is not defined. Does it include the nuclear signal?

As described above, we replaced all measurements using the above animal vs. vegetal cortices to avoid confusion. The nuclear signal is thus not measured in these analyses.

I'm enthusiastic about the results in Figure 7, but I can't really see them very well. Could you please consider changing the color scheme? For single-color figures, it would be helpful to view them as black on white rather than (for example) blue on black. That change is easily achieved with Fiji.

We revised the Figure as suggested.

Page 3 Results section: "At the time of ACD, Insc recruits Pins/LGN to the cortex through Gαi": I understand this sentence to mean that Gαi is an intermediary protein that Insc uses to recruit Pins/LGN. I think the point should be made more clear. As shown in Figure 1, Insc binds to Pins/LGN directly and interacts with cortical polarity proteins directly. Recruitment therefore doesn't appear to require Gαi, but stable association with the membrane (a subsequent step) probably does. That model is shown and described in Figure 6A.

Thank you for the pointer. We clarified our explanations as suggested.

**Reviewer #2 (Recommendations For The Authors):**
The manuscript addresses an interesting question, and uses elegant genetic approaches associated with imaging analyses to elucidate the molecular mechanisms whereby AGS3 and spindle orientation proteins promote asymmetric divisions and specific developmental programs. This considered, it might be worth clarifying a few aspects of the reported findings.(1) In some experimental settings, the presence of AGS3 mutants exacerbates the AGS3 deletion by MO (Figure 4F). Can the author speculate on what can be the molecular explanation?

Thank you for pointing this out. We speculate that AGS1111 and AGS2222 are unable to keep the auto-inhibited forms since they lack GL3 and GL4 as modeled in Figure 6. AGS-MO reduces the endogenous AGS, which compromises the vegetal polarity. In this embryo, constitutive active AGS likely further randomizes the polarity, as evidenced by AGS-OE results in Fig. S7, resulting in an even worse outcome. We elaborated on this part in the text.

(2) Imaging analyses of Figure 4B-C suggest that the mutant AGS1111 does not localise at the vegetal cortex while AGS2222 does (Fig. 4C). However these mutants induce similar developmental defects (Figure 4F). What could be the reason?

We apologize for the confusion in Fig. 4C. The majority of embryos from both AGS1111 and 2222 groups failed to form micromeres and showed AGS localization across the cortex. Among the dozens we examined, 0 embryos from 1111 and 8 embryos from 2222 developed micromeres. Those 8 embryos still showed vegetal cortical localization, so the proportion appears high in Fig. 4B, yet it reflects the minority in the group. In contrast, Development was scored for all embryos (including those that failed to form micromeres), so the graph demonstrates the majority of embryos. To avoid this confusion, we replaced the old Fig. 4C with a new graph that analyzes the cortical signal levels at the vegetal versus animal cortices.

(3) Figure 7 shows the crosstalk between AGS3 and other asymmetry players including NuMA. Vertebrate and *Drosophila* NuMA are ubiquitously present in tissues and localise to the spindle poles in mitosis. However, in Figures 7A and 7E NuMA seems expressed only in a subset of sea urchin embryonic cells. Is this the case?

As the Reviewer rightfully pointed out, Sea urchin NuMA is also present in all cells and localizes to the spindle (please see Fig. 2 of our previous paper PMID: 31439829). AGS is also slightly localized on the spindles of all cells. However, the PLA signal of AGS and NuMA mostly showed up in the vegetal cortex in this study, suggesting that major crosstalk may occur in the vegetal cortex. This does not rule out the possibility that minor interactions may also occur on the spindle or elsewhere in the cell, which was not quantifiable in this study. We clarified this point in the text.